phytools 2.0: an updated R ecosystem for phylogenetic comparative methods (and other things)

Revell Liam J. liam.revell@umb.edu
1 Department of Biology, University of Massachusetts Boston , Boston , MA , USA
2 Facultad de Ciencias, Universidad Católica de la Santísima Concepción , Concepción , Chile
Lapp Hilmar
Electronic publication date: 2024 Jan 5
Publication date: 2024
Volume: 12
Electronic Location ID: e16505
Received 2023 Mar 10; Accepted 2023 Oct 31
Copyright: ©2024 Revell
Copyright year: 2024
Copyright holder: Revell
License: This is an open access article distributed under the terms of the Creative Commons Attribution License, which permits unrestricted use, distribution, reproduction and adaptation in any medium and for any purpose provided that it is properly attributed. For attribution, the original author(s), title, publication source (PeerJ) and either DOI or URL of the article must be cited.
License URL: https://creativecommons.org/licenses/by/4.0/

Keywords: Phylogenetic comparative methods, Phylogeny, Computational biology

Funding: The National Science Foundation DEB-1350474 DBI-1759940 FONDECYT, Chile 1201869 This work was funded by grants from the National Science Foundation (DEB-1350474, DBI-1759940) and FONDECYT, Chile (1201869). The funders had no role in study design, data collection and analysis, decision to publish, or preparation of the manuscript.

==============================
Phylogenetic comparative methods comprise the general endeavor of using an estimated phylogenetic tree (or set of trees) to make secondary inferences: about trait evolution, diversification dynamics, biogeography, community ecology, and a wide range of other phenomena or processes. Over the past ten years or so, the phytools R package has grown to become an important research tool for phylogenetic comparative analysis. phytools is a diverse contributed R library now consisting of hundreds of different functions covering a variety of methods and purposes in phylogenetic biology. As of the time of writing, phytools included functionality for fitting models of trait evolution, for reconstructing ancestral states, for studying diversification on trees, and for visualizing phylogenies, comparative data, and fitted models, as well numerous other tasks related to phylogenetic biology. Here, I describe some significant features of and recent updates to phytools, while also illustrating several popular workflows of the phytools computational software.

Introduction

Phylogenetic trees are the directed graphs used to represent historical relationships among a set of operational taxa that are thought to have arisen via a process of descent with modification and branching (Felsenstein, 2004; Harmon, 2019). Operational taxa in a reconstructed phylogenetic tree might be gene copies, paralogous and orthologous members of a gene family, viral sequences, whole genomes, human cultural groups, or biological species (Nunn, 2011; Yang, 2014). According to its broadest definition, the phylogenetic comparative method corresponds to the general activity of using a known or (most often) estimated phylogenetic tree to learn something else (apart from the relationships indicated by the tree) about the evolutionary process or past, the contemporary ecology, the biogeographic history, or the origins via diversification, of the particular taxa of our phylogeny (Harvey & Pagel, 1991; Felsenstein, 2004; Nunn, 2011; O’Meara, 2012; Harmon, 2019; Revell & Harmon, 2022).

Phylogenetic comparative methods are not new. Perhaps the most important article in the development of the phylogenetic approach to comparative biology (Felsenstein, 1985) was first authored nearly 40 years ago, and was even the subject of a recent retrospective (Huey, Garland & Turelli, 2019). Nonetheless, it is fair to say that phylogenetic comparative methods have seen a relatively impressive expansion and diversification over the past two decades (e.g., Butler & King, 2004; Felsenstein, 2005; O’Meara et al., 2006; Maddison, Midford & Otto, 2007; Hohenlohe & Arnold, 2008; Revell & Collar, 2009; Morlon, Potts & Plotkin, 2010; Stadler, 2011; Etienne & Haegeman, 2012; Goldberg & Igić, 2012; Beaulieu, O’Meara & Donoghue, 2013; Rabosky, 2014; Uyeda & Harmon, 2014; Beaulieu & O’Meara, 2016; Revell, 2021; MacPherson et al., 2022, and many others; reviewed in O’Meara, 2012; Garamszegi, 2014; Harmon, 2019; Revell & Harmon, 2022). This has included the development of new approaches for studying the generating processes of trees (that is, speciation and extinction), the relationship between phenotypic traits and species diversification, and a range of techniques for investigating heterogeneity in the evolutionary process across the branches and clades of the tree of life (O’Meara, 2012; Harmon, 2019; Revell & Harmon, 2022).

Phylogenetic comparative methods have also begun to be applied extensively outside of their traditional domain of evolutionary research. In particular, phylogenies and the comparative method have made recent appearances in studies on infectious disease epidemiology, virology, cancer biology, sociolinguistics, biological anthropology, molecular genomics, and community ecology, among other disciplines (e.g., Moura et al., 2016; Baele et al., 2018; Bentz et al., 2018; Beale et al., 2019; Bushman, McCormick & Sherrill-Mix, 2019; Sánchez-Busó et al., 2019; Valles-Colomer et al., 2019; Freitas et al., 2020; Jezovit et al., 2020; Blinkhorn & Grove, 2021; McLaughlin et al., 2022; Pepke & Eisenberg, 2022; Pozzi, Voskamp & Kappeler, 2022; Compton et al., 2023; Mifsud et al., 2023; Van Borm et al., 2023, and many others).

The scientific computing environment R (R Core Team, 2023) is widely-used in biological research. One of the major advantages that R provides is that it empowers computational scientists and independent developers to build functionality on top of the basic R platform. This functionality often takes the form of what are called contributed R packages: libraries of related functions built by individuals or research collaboratives not part of the core R development team. The growth of importance of R in phylogenetic biology stems entirely from contributed R package. Among these, the most important core function libraries are ape (Paradis, Claude & Strimmer, 2004; Popescu, Huber & Paradis, 2012; Paradis & Schliep, 2019), geiger (Harmon et al., 2008; Pennell et al., 2014), phangorn (Schliep, 2011), and my package, phytools (Revell, 2012).

phytools is an R function library dedicated primarily to phylogenetic comparative analysis, but also including approaches and methodologies in a range of other domains of phylogenetic biology—especially, but not restricted to, visualization. The original article describing phytools is now more than ten years old, and though I recently published a more comprehensive book on the subject of phylogenetic comparative methods in the R environment (Revell & Harmon, 2022), I nonetheless felt that it was time to provide a briefer (although this article is by no means brief) update of phytools specifically—for the primary scientific literature. This is the main purpose of the current article.

The phytools library has now grown to be very large—consisting of hundreds of functions, a documentation manual that is more than 200 pages in length, and tens of thousands of lines of computer code. As such, I thought it would be most useful to compactly summarize some of the functionality of the phytools R package in a few different areas, but each time provide a small set of more detailed example analytical workflows (computational “vignettes”) for the current 2.0 version of the phytools package. A previous version of this article was posted to the preprint server bioRxiv (https://doi.org/10.1101/2023.03.08.531791).

Overview

The phytools R package contains functionality in a diversity of different research areas of phylogenetics and phylogenetic biology. Rather than attempt a comprehensive survey of this functionality here, what I have elected to do instead is briefly review a smaller number of methodological areas, and then illustrate each of these with multiple analysis workflows—including the corresponding R code that can be used to reproduce the analysis and results presented.

My hope is that this article will serve as more than the typical software note placeholder for phytools, and may instead aid R phylogenetic users, both new and old, to be inspired to apply some of the methodologies illustrated herein to their own questions and data. On the other hand, even though it takes the form of a tutorial or R package vignette, this article is not intended to cover nor fully enumerate the complete range of functionality of the package. For that, I would refer readers to the phytools software documentation, my recent book with Luke Harmon (Revell & Harmon, 2022), and my phytools development blog (http://blog.phytools.org).

Installing and loading phytools

This article assumes that readers already have some familiarity with the R computing environment, and have previously installed contributed R packages. Nonetheless, to get started using phytools, the easiest way to install the package locally is by using the R base function called install.packages (in our case, install.packages(”phytools”)), which will download and install phytools from its CRAN page (CRAN is an acronym for Comprehensive R Archive Network: a network of mirror repositories used both to archive and distribute R and contributed R packages). Readers undertaking phylogenetic analysis in the R environment for the first time will note that when we ask R to install phytools, several other R packages are also downloaded and installed automatically. These are packages upon which phytools depends—meaning that phytools uses one or multiple functions exported by each of these packages in its own internal R code. More will be said later about the dependency relationship between phytools and other packages of the R and R phylogenetic ecosystems.

Having installed phytools, if we would like to proceed and use it in an interactive R session, we would normally load it. Loading an R package simply makes the names of the functions of that package visible and available in our current R session. This can be done using the base R function library.

_______________________ library(phytools) packageVersion("phytools")

## [1] ’2.0.0’

packageVersion tells us which version of phytools we have installed. Readers hoping to follow along should ensure that they have a phytools package version that matches or exceeds the value they see above. The phytools package is now loaded.

Discrete characters

The phytools R library now contains a wide range of different methods and models for the analysis of discrete character evolution on trees. For example, phytools can be used to fit and plot an extended Mk model, the continuous-time Markov chain model usually employed to study discrete character evolution on trees (phytools function fitMk, Lewis, 2001; Harmon, 2019), it can fit Pagel’s correlational binary trait evolution model (fitPagel, Pagel, 1994), it can be used to perform stochastic character mapping and reconstruct ancestral states under the Mk and threshold models (make.simmap, simmap, ancThresh, and ancr, Huelsenbeck, Nielsen & Bollback, 2003; Felsenstein, 2005; Felsenstein, 2012; Revell, 2014a), it can fit a polymorphic trait evolution model (fitpolyMk, Revell & Harmon, 2022), it can fit a hidden-rates model (fitHRM, Beaulieu, O’Meara & Donoghue, 2013), it can compare the rate of discrete character evolution between clades and trees (fitmultiMk and ratebytree, Revell et al., 2018; Revell & Harmon, 2022), and it can simulate discrete character data under multiple models (e.g., sim.Mk, sim.history, sim.multiMk).

In this section, I will illustrate the use of just a few of the different discrete character methods that have been implemented in the phytools software.

Stochastic character mapping

Perhaps the most important and widely-used discrete character analysis of phytools is a popular technique referred to as “stochastic character mapping” (Nielsen, 2002; Huelsenbeck, Nielsen & Bollback, 2003; Bollback, 2006). Stochastic character mapping is a method in which we randomly sample discrete character histories (“stochastic maps”) of our trait on the tree under a specified model. By sampling these character histories from their probability distribution under our trait evolution model, and then integrating over the set of histories that we obtain, stochastic mapping helps us to develop a more complete picture of the evolutionary history of our character trait of interest: in terms of the number and types of evolutionary change the character may have undergone; the marginal probabilities that each node of the tree may have been in each condition of the trait; and the branches of the tree with more or fewer character state changes.

Stochastic mapping in phytools can be undertaken in more than one way. An example of a stochastic character mapping analysis could be to first fit (e.g., using maximum likelihood) the character transition model (a variant of the Mk discrete character evolution model of Lewis, 2001; also see, O’Meara, 2012; Harmon, 2019; Revell & Harmon, 2022), and then proceed to randomly sample a set of perhaps 100 or 1,000 stochastic character histories—each consistent with the phenotypic trait observations that we have for the terminal taxa of our tree, and obtained in proportion to their probability under our fitted model. (Other workflows are also popular and possible to undertake within R. For instance, rather than use a single, fixed model of character evolution that has been optimized using Maximum Likelihood, one might instead sample parameters of the evolutionary process from their joint posterior probability distribution using Bayesian MCMC. See Revell & Harmon, 2022 for more details.)

To illustrate stochastic mapping here, I will use a discretely-valued, ecological trait for a small phylogeny of centrarchid fishes from Near, Bolnick & Wainwright (2005; also see (Revell & Collar, 2009; Revell, Toyama & Mahler, 2022). Since the trait (which we will refer to as “feeding mode”) is binary, meaning that it only takes two levels, there are a total of four possible discrete character (extended Mk, see Harmon, 2019) models: equal back-and-forth transitions between the two character values; different rates; and then the two different irreversible trait evolution models.

phytools now allows us to fit a single model or any arbitrary set of models, compare them (if applicable), and pass the model weights and fitted models directly to our stochastic mapping function. If the input is a set of models, as it will be in our example below, our function (called simmap) will then proceed to automatically sample stochastic character histories with probabilities that are proportional to each model weight. Experienced phytools users may figure out that simmap is just a sophisticated wrapper function of make.simmap—the traditional method used for undertaking stochastic character mapping in phytools. A major advantage of sampling stochastic maps across a set of models, rather than under our single best model, is that it allows us to integrate over model uncertainty in direct proportion to the weight of evidence favoring each model in our set.

For this example, and all subsequent examples of the article, our data have been packaged with the phytools library—so we can easily load them in an interactive R session using the base R data function, as follows.

__________________________________________________________________ data(sunfish.tree) data(sunfish.data)

For our Mk model-fitter (which here will be the phytools function fitMk), and for the other discrete character methods of the phytools R package, our input phenotypic trait data typically take the form of a character or factor vector. Personally, I prefer to use factors, because in that case we can more easily access the levels assumed by the character through a call of the base R function levels. This can be very handy.

In this example our input data consists of a data frame in which the feeding.mode column is already coded as a factor. In general, however, had we read this data from an input text file in, for example, comma-separated-value format, R would have created a character (rather than factor) formatted column by default. To adjust this we can set the argument stringsAsFactors=TRUE in our file-reading function, which, in that case, might be the base R function read.csv.

__________________________________________________________________ sunfish.feed mode<-setNames(sunfish.data$feeding.mode,   rownames(sunfish.data)) levels(sunfish.feed_mode)

## [1] "non"  "pisc"

Here we see that our factor vector has two levels: “non” and “pisc”. These two character levels refer to non-piscivorous and piscivorous fishes. Since R factors have no particular character limit on their levels, let us update our data to use these more descriptive names: once again using the function levels. levels is an odd R method in that it can serve both as an extractor function, that pulls out the levels of a factor –as well as acting as an assignment or replacement function, in which the levels of the factor are updated. When we adjust our factor levels for sunfish.feed_mode, we are using levels in this latter fashion.

__________________________________________________________________ levels(sunfish.feed mode)<-c("non-piscivorous",   "piscivorous") levels(sunfish.feed_mode)

## [1] "non-piscivorous" "piscivorous"

Now we are ready to proceed and fit our models. To do so, I will use the phytools function fitMk and fit a total of four models, as previously indicated: “ER”, the equal rates model; “ARD”, the all-rates-different model; and, lastly, the two different irreversible models –one in which non-piscivory can evolve to piscivory, but not the reverse; and a second in which precisely the opposite is true.

For these latter two irreversible models, we will tell fitMk how to build the model by creating and supplying what I will refer to as a “design matrix” for each model that we want to fit. This design matrix should be of dimensions k × k, for k levels of the trait, with integer values in the positions of the matrix corresponding to allowed transitions, and zeros elsewhere. We use different non-zero integer value for each rate that we want to permit to assume a different value in our fitted model. Since our k = 2, this is very easy; however, the same principle would apply to any value of k. (See Revell & Harmon, 2022 for more complex examples.)

__________________________________________________________________ sunfish.ER model<-fitMk(sunfish.tree,sunfish.feed mode,   model="ER") sunfish.ARD_model<-fitMk(sunfish.tree,sunfish.feed_mode,   model="ARD") sunfish.Irr1_model<-fitMk(sunfish.tree,   sunfish.feed_mode,model=matrix(c(0,1,0,0),2,2,   byrow=TRUE)) sunfish.Irr2_model<-fitMk(sunfish.tree,   sunfish.feed_mode,model=matrix(c(0,0,1,0),2,2,   byrow=TRUE))

Having fit our four models, we can also compare them to see which is best-supported by our data. To accomplish this I will use a generic anova function call. anova will print the results of our model comparison; however, it is important that we also assign the value returned by anova to a new object. In my example, I will call this object sunfish.aov—but the name is arbitrary.

__________________________________________________________________ sunfish.aov<-anova(sunfish.ER model,sunfish.Irr1 model,   sunfish.Irr2_model,sunfish.ARD_model)

##                       log(L) d.f.      AIC    weight ## sunfish.ER_model   -13.07453    1 28.14906 0.3486479 ## sunfish.Irr1_model -12.98820    1 27.97640 0.3800846 ## sunfish.Irr2_model -14.20032    1 30.40064 0.1130998 ## sunfish.ARD_model  -12.86494    2 29.72987 0.1581677

This table shown above gives each of our fitted model names, their log-likelihoods, the number of parameters estimated, a value of the Akaike information criterion (AIC), and the Akaike model weights. Smaller values of AIC indicate better support for the corresponding model—taking into account its parameter complexity (Burnham & Anderson, 2003). Model weights can be interpreted as the “weight of evidence” favoring each of our four trait evolution hypotheses (or even the probability that the model is true, given that all possible models are in our set, e.g., Link & Barker, 2006).

Based on this analysis, we might conclude that the first irreversible model (Irr1.model), in which non-piscivory can evolve to piscivory, but not the reverse, is best supported; however, we have a very similar weight of evidence favoring the equal-rates model (ER.model), in which backward and forward transition rates between the two states are identical.

With the result of our anova call in hand (as the sunfish.aov object), we are ready to pass it on directly to phytools’ new generic simmap method. By design, doing so will tell simmap to generate stochastic character maps under each of our four models with relative frequencies that are equal to the weight of evidence supporting of each model.

Here, I will choose to sample 1,000 stochastic character maps—however, this number is somewhat arbitrary. How many is enough? Certainly one or ten are too few, and perhaps a good rule of thumb might be to ask ourselves if we are interested in trait histories that might be expected to be observed (under our model or models) in fewer than one of 100 or 1,000 realizations of the evolutionary process on our phylogeny. If not, then 100 or 1,000 stochastic maps may be enough. There is no harm in generating more, but this can require significant computational effort (depending on the size of our tree), and many empirical studies use a number of stochastic character histories that ranges on this 100–1,000 interval.

__________________________________________________________________ sunfish.simmap<-simmap(sunfish.aov,nsim=1000) sunfish.simmap

## 1000 phylogenetic trees with mapped discrete characters

If we preferred, we could have generated stochastic character maps for just the best-supported of our four models. Using the simmap generic method, this would be done either by supplying our anova result and setting the optional argument weighted=FALSE—or simply by passing our favored Mk model directly to the function.

In spite of the significant number of stochastic simulations involved, this analysis should run fairly quickly (obviously, depending on the speed of our computer). In part this is because we saved computation time by circumventing the need to re-estimate our Mk transition matrix, Q, separately for each sampled model. An additional advantage of this approach is that it has also allowed us to (partly) account for variation in our modeled process of evolution that has due to uncertainty in model selection.

Figure 1 shows a set of six, randomly chosen stochastic character histories for our trait (feeding mode) on our input tree. Readers should see that each of these are consistent with our observed value of the binary trait at the tips of the tree, but that each one differs from the others in the specific hypothesis of trait evolution that it represents.

Figure 1 Six randomly chosen stochastic character maps of feeding mode (non-piscivorous, in dark blue, vs. piscivorous) on a phylogeny of 28 centrarchid fish species.

Stochastic character mapping involves randomly sampling character histories that are consistent with our tip data in proportion to their probability under a model. In this case, histories were sampled under the set of four alternative Mk models of a binary trait, with relative frequencies proportional to the weight of evidence supporting each model. Data are from Near, Bolnick & Wainwright (2005), Revell & Collar (2009), and Revell, Toyama & Mahler (2022). See main text for additional details.

__________________________________________________________________ cols<-setNames(viridisLite::viridis(n=2),   levels(sunfish.feed_mode)) par(mfrow=c(2,3)) plot(sample(sunfish.simmap,6),ftype="i",fsize=0.6,   colors=cols,offset=0.2)

To create my color palette for plotting I used another contributed R package that we have not seen yet called viridisLite by Garnier et al. (2022). viridisLite implements a color palette (known as the “viridis” palette and originally devised by van der Walt and Smith, 2015) that was designed to be both attractive and colorblind-friendly. To replicate Fig. 1 exactly, users should first install viridisLite from CRAN by running install.packages(”viridisLite”) –but they do not need to load it. Calling the contributed package function using the double colon syntax, ::, takes care of that (i.e., viridisLite::viridis).

Although Fig. 1 already gives us a general sense of the uncertainty of our ancestral character history on the tree for our trait, most commonly we do not want to simply graph a subset (or all) of our stochastically mapped trees. Typically, instead, we would first summarize our stochastic character maps (in multiple ways), and then proceed to plot or analyze these summarized findings.

Often, phytools users undertaking stochastic character mapping will compute the posterior probabilities of each value of the character trait at each internal node of the tree, which one can obtain by simply counting the fraction of stochastic maps for which each node is in each of the observed states of our character trait. These values correspond to a form of ancestral state estimation, giving us an approximation of the marginal probabilities that each hypothetical ancestor at each node of the tree was in each of our observed states. We have conditioned on our transition model and its Maximum Likelihood parameter estimates—although in this instance we also integrate across a set of four evolutionary models in proportion to the weight of evidence in support of each one. In phytools, these marginal posterior probabilities values can be obtained using the generic summary method for our object class, which is then easily plotted as follows.

__________________________________________________________________ plot(summary(sunfish.simmap),ftype="i",fsize=0.7,   colors=cols,cex=c(0.6,0.3)) legend("topleft",levels(sunfish.feed_mode),pch=21,   pt.cex=1.5,pt.bg=cols,bty="n",cex=0.8)

A correct interpretation of the graph of Fig. 2 is that it shows the observed discrete character states (at the tips of the tree) and the posterior probabilities from stochastic mapping that each internal node is in each state–all while integrating over our four different transition models in proportion to the weight of evidence for each model.

Figure 2 Posterior probabilities at each ancestral node of the centrarchid tree of Fig. 1 from stochastic character mapping using model weights to sample across four different extended Mk trait evolution models.

See main text for more details.

In addition to node probabilities, phytools users undertaking a stochastic character mapping analysis are often interested in the number of changes of each type that are implied by the evolutionary process and our data. The procedure of stochastic mapping samples full character histories (not just states or probabilities at nodes) and can thus be deployed to produce estimates of the posterior probability distribution of the character changes of each type on specific edges, in specific clades, or across the entire phylogeny, conditioning on our sampled model or models.

To obtain these distributions, we will first call the generic method density which (when applied to an object from stochastic mapping) computes the relative frequency distribution of changes of each type over the whole tree. We can then proceed to graph our results using a different generic plot method, as follows. Remember, our character is binary, so there are only two types of character state changes: from non-piscivorous → piscivorous, and the reverse.

__________________________________________________________________ sunfish.density<-density(sunfish.simmap) sunfish.density

## ## Distribution of changes from stochastic mapping: ##  non-piscivorous->piscivorous        piscivorous->non-piscivorous ##  Min.   :0   Min.   :0 ##  Median :5   Median :2 ##  Mean   :4.13    Mean   :2.24 ##  Max.   :10  Max.   :9 ## ## 95% HPD interval(non-piscivorous->piscivorous): [0, 8] ## 95% HPD interval(piscivorous->non-piscivorous): [0, 6]

__________________________________________________________________ par(mfrow=c(1,2),las=1,cex.axis=0.7,cex.lab=0.8) COLS<-setNames(cols[2:1],sunfish.density$trans) plot(sunfish.density,ylim=c(0,0.6),   transition=names(COLS)[1],colors=COLS[1],main="") mtext("a) transitions to piscivory",line=1,adj=0,   cex=0.8) plot(sunfish.density,ylim=c(0,0.6),   transition=names(COLS)[2],colors=COLS[2],main="") mtext("b) transitions to non-piscivory",line=1,adj=0,   cex=0.8)

The distributions shown in Fig. 3 give the relative frequencies of changes of each type across our set of mapped histories, as well as Bayesian 95% high probability density (HPD) intervals calculated using the R pakage coda (Plummer et al., 2006). For a binary trait like that of this example (and thus with only two types of transitions), we could have instead overlain the distributions of backwards and forwards transitions in character state in a single plot panel. In this particular instance, however, I found that overplotting the two different distributions resulted in a figure that was too difficult to read, and preferred instead to show the distributions in separate panels as in Fig. 3. For multistate characters with more than two types of changes between states, the same plot method will produce a k × k matrix of figure panels, each i, jth panel of which will contain the posterior distribution of changes from character state i to j.

Figure 3 Posterior probability distributions of changes from either (A) non-piscivory to piscivory, or (B) piscivory to non-piscivory, obtained from an analysis of stochastic mapping.

HPD indicates the 95% high probability density interval for changes of each type. See main text for additional details.

An interesting attribute of the character state change distributions for this centrarchid feeding mode analysis is that they are both markedly bi-modal. This is due, in part, to our specific procedure of model-averaging in which we sampled both reversible and irreversible character evolution models in proportion to their weights, and is not something we would have seen had we chosen to analyze just one model or the other. (Recall that the weight of evidence was highly similar between our equal-rates model and the irreversible model in which piscivory is acquired from non-piscivory, but never the reverse. See above.) This pattern is also appropriately captured by the broad HPD intervals on each of the two types of transitions.

Lastly, in addition to these analyses, phytools also makes it quite straightforward to visualize the posterior probabilities of each of the two trait conditions not only at nodes, but also along the branches of the phylogeny. This is accomplished using the phytools function densityMap (Revell, 2013), which creates a graph showing the probability density of stochastic histories in each of our mapped states. By design, in phytools this object can be first created (using the densityMap function), updated (using the method setMap to adjust our color palette for plotting), and then graphed (using a generic plot method that was created for this specific object class). I will illustrate this set of procedures in the following code block. The resultant plot is shown in Fig. 4.

Figure 4 Posterior probability density of each of the two character levels, piscivory and non-piscivory, based on stochastic character mapping, graphed along the edges of a tree of centrarchid fishes using a color gradient.

See main text for more details.

__________________________________________________________________ sunfish.densityMap<-densityMap(sunfish.simmap,plot=FALSE,   res=1000) sunfish.densityMap

## Object of class "densityMap" containing: ## ## (1) A phylogenetic tree with 28 tips and 27 internal nodes. ## ## (2) The mapped posterior density of a discrete binary character ##     with states (non-piscivorous, piscivorous).

__________________________________________________________________ sunfish.densityMap<-setMap(sunfish.densityMap,   viridisLite::viridis(n=10)) plot(sunfish.densityMap,lwd=3,outline=TRUE,   fsize=c(0.6,0.7),legend=0.1)

Having enthusiastically demonstrated the model-averaging feature of the new phytools simmap method, I would be remiss if I failed to note that this is not (as yet) the standard workflow for ancestral state reconstruction of discrete characters in general, nor for stochastic mapping in particular. More typically, researchers select the best model and then proceed to hold this model (and its parameters) constant through subsequent calculations (e.g., Yang, 2014), or they sample parameter values for a single model from their joint posterior distribution using MCMC (e.g., shown in Revell & Harmon, 2022). I think, however, that there is a very strong case to be made that if, for example, 51% of the weight of evidence points to a model in which a specific node has a high conditional probability of being in state a, while 49% of the weight of evidence points to a model wherein the same node has a high probability of being in state b, then the correct marginal probability that the node is actually in state a is probably closer to 0.5 than 1.0. Indeed, this would be our exact interpretation of this result if we consider the model weights as the probability that each model is correct (assuming that all possible models are in our set, e.g., Link & Barker, 2006).

Apart from the analyses shown, stochastic mapping as implemented in phytools is a very flexible method via which we might sample the matrix of transition rates from its posterior distribution under a model, incorporate uncertainty in the character state values for different species, take into account polymorphic character conditions and hidden-rates of trait evolution, and integrate over phylogenetic uncertainty. A comprehensive survey of this functionality is beyond the scope of the present article; however, considerable additional information about stochastic mapping in R can be found in the phytools documentation pages as well as elsewhere online.

The polymorphic trait evolution model

Another important, but much more recently-added, tool in the phytools R package is a method (called fitpolyMk) that is designed to fit a discrete character evolution model to trait data containing intraspecific polymorphism (Revell & Harmon, 2022). In this case, our model is one in which an evolutionary transition from (say) character state a to character state b must first pass through the intermediate polymorphic condition of a + b. This model starts off very simply –but will become increasingly complicated for increasing numbers of monomorphic conditions of our trait. Not only that, but as soon as we have more than two monomorphic states, we must also consider whether our character is evolving in an ordered (Fig. 5A) or unordered (Fig. 5B) fashion (Revell & Harmon, 2022). Figure 5 shows the general structure of an ordered and unordered polymorphic trait evolution model—both for the same, underlying number of monomorphic conditions of our trait (four).

Figure 5 Example structures of two alternative polymorphic trait evolution models for characters with four monomorphic conditions: (A) an ordered model with states 0 to 3; (B) an unordered model, with states a, b, c, and d.

The maximum parameter complexity of each model corresponds to 2× the number of double-ended arrows in the panel. See main text for additional details.

__________________________________________________________________ par(mfrow=c(1,2)) graph.polyMk(k=4,ordered=TRUE,states=0:3,   mar=rep(0.1,4)) mtext("a) ordered polymorphic model",line=-1,adj=0.2,   cex=0.8) graph.polyMk(k=4,ordered=FALSE,states=letters[1:4],   mar=rep(0.1,4),spacer=0.15) mtext("b) unordered polymorphic model",line=-1,   adj=0.2,cex=0.8)

Obviously, the potential parameter complexity of the unordered polymorphic trait evolution model is higher than the ordered model. Since there exists an unordered model that also has all ordered models as a special case, ordered and unordered models can be compared using likelihood-ratio tests (if nested) or information criteria.

To try out our polymorphic trait evolution model, let us use an excellent, recently-published dataset from Halali et al. (2020) consisting of a phylogenetic tree containing 287 Mycalesina butterfly species and data for butterfly habitat use. Halali et al. (2020) coded habitat as a polymorphic trait in which, for example, a species using both “forest” and forest “fringe” habitat would be recorded as ”forest+fringe”. In this case, our polymorphic trait evolution model will assume that to evolve from forest specialization to fringe specialization, a species must first (at least transiently) evolve through the polymorphic condition of using both habitats at once. This seems logical.

The Halali et al. (2020) dataset and tree now come packaged with the phytools library, so both can be loaded using the data function, just as we saw for the centrarchid data and tree of our previous example.

__________________________________________________________________ data(butterfly.tree) data(butterfly.data)

Let us begin by inspecting our data.

__________________________________________________________________ head(butterfly.data)

##                                     habitat ## Myc_francisca_formosana? forest+fringe+open ## Bic_cooksoni                           open ## Bic_brunnea                          forest ## Bic_jefferyi                    fringe+open ## Bic_auricruda_fulgida                forest ## Bic_smithi_smithi             forest+fringe

fitpolyMk requires us to separate the different states in each polymorphic condition using the + symbol, but does not demand that our states be ordered in a consistent manner. In other words, a+b and b+a would be considered (properly) to be same polymorphic condition. As a first preliminary step in our analysis, we can proceed to extract the column of habitat use data (habitat in our data frame) as a vector, and then print the different levels that it takes.

__________________________________________________________________ butterfly.habitat<-setNames(butterfly.data$habitat,   rownames(butterfly.data)) print(levels(butterfly.habitat))

## [1] "forest"             "forest+fringe"      "forest+fringe+open" ## [4] "fringe"             "fringe+open"        "open"

Now, let us proceed to fit our polymorphic trait evolution model to these data. In this instance, I will fit a grand total of six different models. This is not a comprehensive set of the conceivable models for polymorphic data with these levels, but it seemed like a reasonable selection for illustrative purposes.

The first three of these models all suppose that the evolution of my discrete character is totally unordered. Among this set, we will imagine, first, equal transition rates between all monomorphic states or polymorphic conditions. For our second model, we will permit all possible transition rates between states or state combinations to assume different values. Finally, for our third model we will assume that the acquisition of polymorphism (or its increase) occurs with one rate, whereas the loss (or decrease) of polymorphism occurs with another, separate rate. We refer to this last scenario as the “transient model” following Revell & Harmon (2022). This name for the model comes from the general notion that if the rate of loss exceeds the rate of gain, then polymorphism will typically be relatively transient in nature. Since polymorphism tends to be less frequently observed in the types of data that typify many phylogenetic comparative studies, including this model in our set seems like a reasonable idea.

To get our remaining three models, and reach the six total models that I promised at the outset of this section—for each of the three listed above in which character evolution is unordered, we will simply add a second ordered model in which we assume that character evolution for our three monomorphic conditions tends to proceed as follows: forest ↔ fringe ↔open –not forgetting, of course, about the intermediate polymorphic conditions found between each pair of monomorphic states.

To fit our first three models in R, we will use the function fitpolyMk from the phytools package as follows.

__________________________________________________________________ butterfly.ER unordered<-fitpolyMk(butterfly.tree,   butterfly.habitat,model="ER")

## ## This is the design matrix of the fitted model. ## Does it make sense? ## ##                    forest fringe open ## forest                  0      0    0 ## fringe                  0      0    0 ## open                    0      0    0 ## forest+fringe           1      1    0 ## forest+open             1      0    1 ## fringe+open             0      1    1 ## forest+fringe+open      0      0    0 ##                    forest+fringe forest+open fringe+open ## forest                         1           1           0 ## fringe                         1           0           1 ## open                           0           1           1 ## forest+fringe                  0           0           0 ## forest+open                    0           0           0 ## fringe+open                    0           0           0 ## forest+fringe+open             1           1           1 ##                    forest+fringe+open ## forest                              0 ## fringe                              0 ## open                                0 ## forest+fringe                       1 ## forest+open                         1 ## fringe+open                         1 ## forest+fringe+open                  0

By default, fitpolyMk begins by printing out the design matrix of the model for us to verify. The design matrix is of dimensions dictated by the number of states and polymorphic conditions of our character, with integers populating the different types of transitions, from row to column, that should be permitted under our model—and zeros indicating disallowed transition types. The specific integer values do not mean anything; however, different integer values imply that the corresponding transitions will be allowed to take place with different rates under our model.

This can be helpful, because we should find that it corresponds with the design matrix that was discussed under the simpler Mk model of the previous section—as well as with the graphed models of Fig. 5. If we do not want the design matrix to print, though, we can turn off this behavior simply by setting the optional argument quiet=TRUE. Let us do that for our remaining two unordered models.

__________________________________________________________________ butterfly.ARD unordered<-fitpolyMk(butterfly.tree,   butterfly.habitat,model="ARD",quiet=TRUE,   opt.method="optimParallel",rand_start=TRUE) butterfly.transient_unordered<-fitpolyMk(   butterfly.tree,butterfly.habitat,   model="transient",quiet=TRUE,   opt.method="optimParallel",rand_start=TRUE)

Astute readers may notice that I added two additional arguments that did not feature in my previous fitpolyMk function call: opt.method=“optimParallel” and rand_start=TRUE. The former tells my optimizer to use the optimParallel package (Gerber & Furrer, 2019) for optimization. The latter says “choose random starting values.” Both of these, and sometimes multiple optimization replicates, may be required to find our Maximum Likelihood solution for these complex models. In fact, I virtually guarantee it.

Now we can proceed to do the same thing, but this time updating the argument value ordered to ordered=TRUE. When we switch from fitting an unordered polymorphic trait evolution model to our ordered model, it suddenly becomes critical that we specify the order levels using the optional function argument order. If order is not indicated, fitpolyMk will simply assume that our characters are ordered alphanumerically –but this is very rarely likely to be correct. (By chance, it happens to be true of our butterfly dataset. I assigned the argument order anyway, just to be safe.)

__________________________________________________________________ levs<-c("forest","fringe","open") levs

## [1] "forest" "fringe" "open"

__________________________________________________________________ butterfly.ER ordered<-fitpolyMk(butterfly.tree,   butterfly.habitat,model="ER",ordered=TRUE,order=levs,   quiet=TRUE) butterfly.ARD_ordered<-fitpolyMk(butterfly.tree,   butterfly.habitat,model="ARD",ordered=TRUE,   order=levs,quiet=TRUE,opt.method="optimParallel",   rand_start=TRUE) butterfly.transient_ordered<-fitpolyMk(butterfly.tree,   butterfly.habitat,model="transient",ordered=TRUE,   order=levs,quiet=TRUE,opt.method="optimParallel",   rand_start=TRUE)

Now, with all six models in hand, let us compare them using an anova call as follows. I will save my results from our model comparison to the object butterfly.aov.

__________________________________________________________________ butterfly.aov<-anova(butterfly.ER ordered,   butterfly.ER_unordered,   butterfly.transient_ordered,   butterfly.transient_unordered,   butterfly.ARD_ordered,   butterfly.ARD_unordered)

##                                  log(L) d.f.      AIC       weight ## object                        -329.0390    1 660.0779 1.472873e-09 ## butterfly.ER_unordered        -355.8122    1 713.6244 3.472845e-21 ## butterfly.transient_ordered   -329.0205    2 662.0409 5.519508e-10 ## butterfly.transient_unordered -353.4496    2 710.8991 1.356691e-20 ## butterfly.ARD_ordered         -297.7376   12 619.4753 9.658773e-01 ## butterfly.ARD_unordered       -295.0807   18 626.1614 3.412273e-02

A quick word of caution to readers is probably merited here. These models can be quite difficult to optimize, meaning that it is not inconceivable to imagine that (in spite of our best efforts) fitpolyMk has not converged on the true Maximum Likelihood solution for one model or another. Although the true best solution may be unknowable (this is why we use numerical optimization to try and ascertain it), common sense can be a valuable defense against very obvious failures of optimization. For instance, had we found that the most complex model (in our case, butterfly.ARD_unordered) had a lower likelihood than any of its nested counterparts (for instance, butterfly.ARD_ordered), this would give us very strong cause to believe that one or both models had not converged, and that we should perhaps try different random starts or alternative optimization routines to try to find better solutions.

Nonetheless, taking our fitted models at face value, model comparison shows that (among the models in our set) the best supported by far (accounting for parameter complexity) is the ordered, all-rates-different model. phytools has a function to graph this model, so let us go ahead and use it (Fig. 6).

Figure 6 Best-fitting polymorphic trait evolution model for the evolution of habitat use in Mycalesina butterflies.

Data and phylogeny are from Halali et al. (2020). See main text for more details.

__________________________________________________________________ plot(butterfly.ARD ordered,asp=0.65,mar=rep(0.1,4),   cex.traits=0.8) legend("bottomleft",legend=c(paste("log(L) =",   round(logLik(butterfly.ARD_ordered),2)),   paste("AIC =",round(AIC(butterfly.ARD_ordered),2))),   bty="n",cex=0.8)

Just as with our fitted Mk models from the prior section, we can also pass this model object to our generic stochastic character mapping method, simmap. When we do, simmap will automatically generate a set of 100 stochastic character maps under our fitted model. We could have likewise passed simmap our anova results, just as we did with our “fitMk” objects in the centrarchid example, above. In this case, however, nearly all the weight of evidence fell on one model, so this wouldn’t really make much difference anyway.

__________________________________________________________________ butterfly.simmap<-simmap(butterfly.ARD ordered) butterfly.simmap

## 100 phylogenetic trees with mapped discrete characters

Now that we have our stochastically mapped trees, let us compute a summary, just as we did in the prior section.

__________________________________________________________________ butterfly.summary<-summary(butterfly.simmap)

Much as we saw earlier, the object from our generic summary call can be conveniently plotted using phytools. In this case, rather than using the viridis palette we saw earlier, I will use the base graphics function rgb to attempt to select colors for plotting that are evenly spaced in a red-green-blue color space in which the “corners” (red, green, and blue) correspond to the three monomorphic states of our data. Does that make sense? I am colorblind, so it is hard for me to be sure how the rgb color space captures the “intermediacy” of the polymorphic conditions between the corresponding monomorphic states. Nonetheless, I hope the reader can use this demonstration as an example of how to specify custom palettes, rather than an endorsement of a specific palette.

__________________________________________________________________ hab.cols<-setNames(c(rgb(0,1,0),rgb(0,0.5,0.5),   rgb(1/3,1/3,1/3),rgb(0,0,1),rgb(0.5,0.5,0),   rgb(1,0,0)),levels(butterfly.habitat)) par(fg="transparent") h<-max(nodeHeights(butterfly.tree)) plot(butterfly.summary,type="arc",ftype="off",   colors=hab.cols,cex=c(0.4,0.2),part=0.5,lwd=1,   arc_height=0.4,ylim=c(-3,35)) par(fg="black") legend("topleft",names(hab.cols),pch=21,pt.bg=hab.cols,   pt.cex=1.5,cex=0.8,bty="n") axis(1,pos=-1,at=h-seq(0,h,by=5)+0.4*h,   labels=seq(0,h,by=5),cex.axis=0.8) axis(1,pos=-1,at=-h+seq(0,h,by=5)-0.4*h,   labels=seq(0,h,by=5),cex.axis=0.8)

Excellent. Figure 7 shows both the observed (at the tips) and reconstructed (at the internal nodes) marginal posterior probabilities for each of our states and polymorphic conditions.

Figure 7 Posterior probabilities of monomorphic or polymorphic conditions at internal nodes from stochastic mapping under an ordered, ARD polymorphic model of trait evolution.

Data and phylogeny are from Halali et al. (2020). The horizontal axis is in millions of years before the present. See main text for additional details.

Lastly, let us graph the posterior distribution of the accumulation of lineages in each state over time, using the phytools function ltt as follows. We can even do this while retaining the same color palette as we used for Fig. 7. (We will learn more about ltt in a subsequent section.) The resultant plot in Fig. 8 simultaneously shows not only the accumulation of lineages in each mono- or polymorphic state, but also the variation attributable to uncertainty in the evolutionary history of our group from our stochastic character maps. Even though Fig. 8 looks very cool—to be fair, this type of graph is only especially meaningful for the situation in which the taxa of our phylogeny have been completely or close to completely sampled. In this example, we have around 85% of described species for the group (Halali et al., 2020)—a high enough sampling fraction, perhaps, to make this plot meaningful. Sampling fractions in phylogenetic comparative biology, however, are often much lower.

Figure 8 Lineage-through-time plot showing the reconstructed accumulation of lineages in each polymorphic condition or monomorphic state over time, from 100 stochastic character maps.

Data and phylogeny are from Halali et al. (2020). See main text for additional details.

__________________________________________________________________ butterfly.ltt<-ltt(butterfly.simmap) par(mar=c(4.1,4.1,1.1,1.1)) ave_butterfly.ltt<-plot(butterfly.ltt,show.total=FALSE,   bty="n",las=1,cex.axis=0.7,cex.lab=0.8,colors=hab.cols,   legend=FALSE,xlim=c(0,1.05*max(nodeHeights(butterfly.tree)))) k<-length(levels(butterfly.habitat)) legend("topleft",paste(1:k,". ",levels(butterfly.habitat),sep=""),   bty="n",pch=22,pt.bg=hab.cols,pt.cex=1.2,cex=0.7) nn<-length(ave_butterfly.ltt$times) text(x=rep(ave_butterfly.ltt$times[nn],k),   y=ave_butterfly.ltt$ltt[nn,1:k],   labels=paste(1:k,".",sep=""),pos=4,cex=0.7)

As with stochastic mapping under the standard Mk model, implementation of the polymorphic trait evolution model in phytools also allows us to take into account uncertainty in the data or in the phylogeny as well as variation in the rate of evolution between different clades and branches of the tree under the hidden rates model of Beaulieu, O’Meara & Donoghue (2013, also see below). Covering all of this functionality here is not possible; however, additional information is available via phytools documentation pages and online.

Hidden rate models

In addition to fitpolyMk, another relatively recent addition to the phytools package for discrete character analysis has been the function fitHRM. fitHRM implements the hidden-rates trait evolution model of Marazzi et al. (2012) and Beaulieu, O’Meara & Donoghue (2013). Under this model, which is closely related to the covarion model from phylogenetic inference (Galtier, 2001; Penny et al., 2001), each observed state of our discrete trait may have one or more unobserved levels. These different hidden trait levels are each free, in turn, to possess different rates of transition to the other observed character conditions in our trait space. An important aspect of this model is that it allows us to explicitly capture heterogeneity in the evolutionary process of trait evolution—not only between different observed conditions of our character, but also across different branches and clades of the phylogeny (e.g., Beaulieu, O’Meara & Donoghue, 2013; King & Lee, 2015). Note that both hidden-rate models and ancestral character estimation, which we will see more of below, are also implemented in the excellent corHMM package of Beaulieu et al. (2022).

To illustrate use of the hidden-rates model in phytools, we can load a phylogenetic tree of lizards from the diverse South American family Liolaemidae, along with a dataset for parity mode (oviparity vs. viviparity) and different environmental trait measures. Both the phylogeny and the trait data were obtained from Esquerré et al. (2019) and, like the other datasets used in this article, are now packaged with the phytools R library.

__________________________________________________________________ data(liolaemid.tree) data(liolaemid.data)

We can start by inspecting our data object.

__________________________________________________________________ head(liolaemid.data)

##                         parity_mode max_altitude temperature ## Ctenoblepharys_adspersa           O          750       23.05 ## Liolaemus_abaucan                 O         2600       20.20 ## Liolaemus_albiceps                V         4020       12.38 ## Liolaemus_andinus                 V         4900       11.40 ## Liolaemus_annectens               V         4688        5.10 ## Liolaemus_anomalus                O         1400       23.78

We should see that the two levels of our discrete character of interest, parity mode, have been coded as “O” (oviparity) and “V” (viviparity), respectively. To proceed and use fitHRM to fit hidden-rate models with phytools, we must next extract the parity mode of our liolaemid species. An easy way to do that, as we have seen in prior sections, is via the handy function setNames.

__________________________________________________________________ liolaemid.parity<-setNames(liolaemid.data$parity mode,   rownames(liolaemid.data))

One flavor of hidden-rates model, as described in Revell & Harmon (2022), in which we call it the “umbral” model, from umbral meaning threshold in Spanish), allows transitions only between specific, labile conditions of the trait. Transitions in observed state are not permitted, on the other hand, any time a lineage finds itself in the hidden, inert level. (This model is also closely related to what was referred to as the “precursor model” by Marazzi et al., 2012.) Let us try to fit this model to our data using two rate categories per observed state of our character. This is specified using the function argument ncat =2. (We could have chosen to model more than two levels per observed trait value, or even a different number of levels for the “O” and “V” conditions, respectively.)

Since this model class can be quite difficult to fit to data, fitHRM is designed to use multiple optimization iterations (10 by default, but this can be adjusted by modifying the optional function argument niter) with different random starting values. These optimization iterations can also be parallelized across our computer cores by specifying parallel=TRUE. Just as was true of fitMk and fitpolyMk, optimization in fitHRM can also be parallelized using optimParallel (Gerber & Furrer, 2019)—however, we must not try to set parallel = TRUE and opt.method =“optimParallel” at the same time.

__________________________________________________________________ lliolaemid.hrm<-fitHRM(liolaemid.tree,liolaemid.parity,   ncat=2,umbral=TRUE,pi="fitzjohn",parallel=TRUE)

Does it make sense?    O O* V V* O  0  1 2  0 O* 3  0 0  0 V  4  0 0  5 V* 0  0 6  0 Opened cluster with 10 cores. Running optimization iterations in parallel. Please wait....

Much as we saw with fitpolyMk, by default fitHRM starts by printing the model design matrix to screen for users to inspect. This default setting can be turned off using quiet=TRUE.

Let us review our fitted model.

__________________________________________________________________ liolaemid.hrm

## Object of class "fitHRM". ## ## Observed states: [ O, V ] ## Number of rate categories per state: [ 2, 2 ] ## ## Fitted (or set) value of Q: ##            O        O*         V       V* ## O  -1.431683  0.000000  1.431683 0.000000 ## O*  0.041029 -0.041029  0.000000 0.000000 ## V   2.267185  0.000000 -2.812138 0.544953 ## V*  0.000000  0.000000  0.000000 0.000000 ## ## Fitted (or set) value of pi: ##  O O*  V V* ##  0  1  0  0 ## due to treating the root prior as (a) nuisance. ## ## Log-likelihood: -59.117373 ## ## Optimization method used was "optim" ## ## R thinks it has found the ML solution.

The structure of the transition matrix Q ought to match our design matrix in that optimized transition rates should only be found in matrix cells populated by non-zero integers in our printed design. (Except for the matrix diagonal which always contains a value equal to the negative row sum, O’Meara, 2012; Revell & Harmon, 2022.) Here we see that it does—although some Maximum Likelihood transition rate values, such as the transition rate from O (the labile condition of oviparity) to O* (the inert condition) are not different from zero in the fitted model (also see Fig. 9).

Figure 9 Marginal ancestral state reconstruction of parity mode (oviparity vs. viviparity) in liolaemid lizards under the hidden-rates model.

Phylogeny and data based on Esquerré et al. (2019). Inset panel shows best-supported hidden-rates model. See main text for additional details.

A conventional analysis workflow would typically involve comparing this fitted model to a standard Mk model (discussed above, also see Harmon, 2019), as well as, perhaps, other variants of the hidden-rates model (Beaulieu, O’Meara & Donoghue, 2013; Revell & Harmon, 2022). Here, I will compare our umbral model to both a standard extended Mk model with different backward and forward rates of transitions (the “ARD” model), as well as to a slightly more complex hidden-rates model in which transitions are allowed between the hidden condition levels, just at different rates. We could fit the Mk model using the phytools function fitMk, as we did earlier—but here I will do it using fitHRM by setting ncat (the number of rate categories for each level of the trait) to ncat=1. This also helps us see that standard Mk models are special cases of the hidden-rates model—just without hidden rate categories.

__________________________________________________________________ liolaemid.mk<-fitHRM(liolaemid.tree,liolaemid.parity,   ncat=1,pi="fitzjohn",parallel=TRUE,quiet=TRUE) liolaemid.full<-fitHRM(liolaemid.tree,liolaemid.parity,   ncat=2,pi="fitzjohn",parallel=TRUE,quiet=TRUE) anova(liolaemid.mk,liolaemid.hrm,liolaemid.full)

##                   log(L) d.f.      AIC     weight ## object         -64.27046    2 132.5409 0.21754700 ## liolaemid.hrm  -59.11737    6 130.2347 0.68917781 ## liolaemid.full -59.11732    8 134.2346 0.09327518

By comparing these three models we see that there is relatively little support for the extended Mk (“ARD”) model and for the full hidden-rates model, compared to our best-supported model: the original, umbral model. Indeed, the full hidden-rates model actually has virtually the same likelihood as our umbral model, but with two additional parameters to be estimated.

phytools now makes it very easy to undertake joint or marginal ancestral state reconstruction (e.g., Yang, 2014; Revell & Harmon, 2022) under a hidden-rate model, as well as under other models we have seen in this article (such as the standard extended Mk model and the polymorphic trait evolution model) via the phytools generic method ancr. Much as with the simmap method described previously, all we need to do is pass our fitted model object to the method, and ancr will do the rest. Although I will not show it here, ancr is also capable of computing model-averaged ancestral states if we simply supply it with a set of models (in lieu of a single model) in the form an object computed using an anova method call. It can also perform joint reconstruction, rather than the marginal ancestral state estimation shown here. (For more information on the difference between marginal and joint ancestral state estimation for discrete characters, see Yang, 2014; Revell & Harmon, 2022.)

__________________________________________________________________ liolaemid.hrm asr<-ancr(liolaemid.hrm,tips=TRUE) print(liolaemid.hrm_asr,printlen=12)

## Marginal ancestral state estimates: ##            O       O*        V V* ## 258 0.000000 1.000000 0.000000  0 ## 259 0.000000 1.000000 0.000000  0 ## 260 0.000000 1.000000 0.000000  0 ## 261 0.000000 1.000000 0.000000  0 ## 262 0.000000 1.000000 0.000000  0 ## 263 0.000000 1.000000 0.000000  0 ## 264 0.000000 1.000000 0.000000  0 ## 265 0.000000 1.000000 0.000000  0 ## 266 0.000025 0.999967 0.000009  0 ## 267 0.005267 0.993303 0.001430  0 ## 268 0.005649 0.992315 0.002037  0 ## 269 0.140376 0.820768 0.038856  0 ## ... ## ## Log-likelihood = -59.117373

Lastly, this marginal ancestral state reconstruction can easily be plotted on the tree using a phytools plot method for the object class. Here, just for fun, I have also inset a visualization of our fitted umbral hidden-rates model. We can see from this best-supported model that although the observed condition of parity mode may not satisfy Dollo’s Law (Lee & Shine, 1998) in liolaemid lizards, under the umbral model parity mode evolution does appear to have a hidden, absorbing (i.e., irreversible) viviparous condition (Fig. 9), from which oviparous reproductive mode can no longer re-evolve.

__________________________________________________________________ cols<-setNames(c("#FFE5B4","#F0EAD6","#E97451",   "#880808"),colnames(liolaemid.hrm_asr$ace)) plot(liolaemid.hrm_asr,legend=FALSE,   args.plotTree=list(type="arc",arc_height=0.5,     fsize=0.25,offset=5,xlim=c(-65,65),ylim=c(0,65)),   args.nodelabels=list(piecol=cols,cex=0.3),   args.tiplabels=list(cex=0.15)) pp<-plot(liolaemid.hrm,add=TRUE,xlim=c(-4,2),   ylim=c(-1.3,4.7),spacer=0.2,offset=0.1) invisible(mapply(plotrix::draw.circle,x=pp$x,y=pp$y,   col=cols,MoreArgs=list(radius=strheight("0"),     border="transparent"))) text(pp$x,pp$y,pp$states,col=c("black","black","white",   "white"))

Continuous characters

Numerous continuous trait methods exist in the phytools package. For example, phytools can be used to measure phylogenetic signal (phylosig, Pagel, 1999; Blomberg, Garland & Ives, 2003; Revell, Harmon & Collar, 2008), it can fit multi-rate Brownian evolution models (brownie.lite, brownieREML, evol.rate.mcmc, multirateBM, ratebytree, and rateshift, O’Meara et al., 2006; Revell et al., 2012; Revell et al., 2018; Revell, 2021; Revell & Harmon, 2022), it can perform phylogenetic canonical correlation and principal components analysis (phyl.cca and phyl.pca, Revell & Harrison, 2008; Revell, 2009), it can reconstruct ancestral states under multiple evolutionary models (anc.Bayes, anc.ML, anc.trend, and fastAnc, Schluter et al., 1997; Revell & Harmon, 2022), it can use continuous trait data to place a fossil or missing lineage into a reconstructed tree (locate.fossil and locate.yeti, Felsenstein, 2002; Revell et al., 2015), it can fit a multivariate Brownian model with multiple evolutionary correlations on the tree (evol.vcv and evolvcv.lite, Revell & Collar, 2009; Revell, Toyama & Mahler, 2022), and it can perform various types of continuous character numerical simulation on phylogenies (e.g., branching.diffusion, fastBM, sim.corrs, sim.rates).

Here I will start by illustrating the measurement of phylogenetic signal (phylosig), then I will demonstrate Bayesian ancestral state estimation (anc.Bayes). I will show how to fit a variable-correlation multivariate Brownian trait evolution model (evolvcv.lite), and, finally, I will demonstrate a relatively new multi-rate trait evolution model that uses the estimation technique of penalized likelihood (multirateBM).

Phylogenetic signal

Perhaps the simplest phylogenetic comparative analysis that we could choose to undertake for a continuous trait data in R is the measurement of phylogenetic signal (Pagel, 1999; Blomberg, Garland & Ives, 2003; Revell, Harmon & Collar, 2008). Phylogenetic signal has been defined in a number of different ways, but could be considered to be the basic tendency of more closely related species to bear more similarity (one to another) than they do to more distant taxa (Revell, Harmon & Collar, 2008). Apart from its definition, phylogenetic signal can likewise be quantified in various manners; however, undoubtedly the two most popular metrics are Blomberg, Garland & Ives (2003) K statistic, and Pagel’s (1999) λ. Conveniently, both of these can be calculated using the phytools package.

To get started in this undertaking, let us load some data from phytools consisting of a phylogenetic tree of elopomorph eels and a data frame of phenotypic traits. Both tree and data were obtained from an article by Collar et al. (2014) and are now packaged with phytools.

__________________________________________________________________ data(eel.tree) data(eel.data) head(eel.data)

##                   feed_mode Max_TL_cm ## Albula_vulpes       suction       104 ## Anguilla_anguilla   suction        50 ## Anguilla_bicolor    suction       120 ## Anguilla_japonica   suction       150 ## Anguilla_rostrata   suction       152 ## Ariosoma_anago      suction        60

Having loaded these data, we will next extract one variable from our data array. Phylogenetic signal can be measured for any continuous trait, so we will use maximum total length: here represented by the column of our data frame called ”Max_TL_cm”. As is often the case, we will transform our data to a log scale. (There are multiple reasons log transformations are favored by comparative biologists working on interspecies data. One is that it makes a, say, 10% change equal, regardless of whether it occurs in an elephant or a mouse. See Revell & Harmon, 2022 for more details.)

__________________________________________________________________ eel.lnTL<-setNames(log(eel.data$Max TL cm),   rownames(eel.data))

Next, we will compute a value of the K statistic of Blomberg, Garland & Ives (2003) using the phytools function phylosig. phylosig calculates K by default (that is, without specifying an argument for method), but if I add the argument value test=TRUE, phylosig will also conduct a statistical test of the measured value of K by comparing it to a null distribution of K obtained by permuting our observed trait values randomly across the tips of the phylogeny.

__________________________________________________________________ eel.Blomberg K<-phylosig(eel.tree,eel.lnTL,test=TRUE) eel.Blomberg_K

## ## Phylogenetic signal K: 0.362879 ## P-value (based on 1000 randomizations): 0.036

K has an expected value of 1.0 under Brownian motion (Blomberg, Garland & Ives, 2003). The lower value that we observe here thus indicates less phylogenetic signal than expected under Brownian evolution; whereas a value higher than 1.0 would’ve indicated more. Our significance test shows us that this value of K, though numerically modest, is nonetheless significantly greater than we would expect to find in data that were entirely random with respect to the tree.

In addition to Blomberg, Garland & Ives (2003) K, phytools also can be used to estimate Pagel’s (1999) λ statistic. λ measures phylogenetic signal as a scalar multiplier of the correlations of related taxa in our tree (Revell & Harmon, 2022). That is to say, if λ has a value less than 1.0, this would indicate that related species in our phylogeny have a lower degree of “autocorrelation” than expected under Brownian evolution. In fact, a value of λ close to zero could be taken to indicate that related species are not phenotypically correlated at all.

We use Maximum Likelihood to find the value of λ that makes our observed data most probable. Since it is straightforward to compute a likelihood for any allowable value of λ, including λ = 0, we can very easily proceed to test a null hypothesis of no phylogenetic signal in our data by simply calculating a likelihood ratio in which we compare λ = 0 to our Maximum Likelihood estimate. Indeed, this is the test performed by phytools if method=“lambda” and test=TRUE.

__________________________________________________________________ eel.Pagel lambda<-phylosig(eel.tree,eel.lnTL,   method="lambda",test=TRUE) eel.Pagel_lambda

## ## Phylogenetic signal lambda: 0.673729 ## logL(lambda): -54.3016 ## LR(lambda=0): 5.18173 ## P-value (based on LR test): 0.0228256

This result tells us that we have found significant phylogenetic signal in our trait by both measures. Although K and λ tend to be correlated, it is entirely possible that we could have found significant K and non-significant λ, or vice versa. This is not a contradiction. The concept of phylogenetic signal is one of phenotypic similarity among related species—but K and λ measure this concept via two entirely different procedures.

Along with the simple calculation of phylogenetic signal, phytools also contains several methods to visualize our results. In particular, for Blomberg, Garland & Ives (2003) K we can plot the permutation distribution of K alongside our observed measure. For Pagel’s λ, we can plot the likelihood surface, our Maximum Likelihood solution, and the likelihood of λ = 0: the null hypothesis of our statistical tests. Both of these plots are illustrated in Fig. 10 for our eel body length data.

Figure 10 (A) Blomberg, Garland & Ives (2003) measured value of the K statistic for phylogenetic signal, compared to a null distribution of K obtained via randomization. (B) Pagel’s (1999) λ statistic for phylogenetic signal, also showing the likelihood surface. Data consist of maximum body length (on a log scale) from 61 species of elopomorph eels (Collar et al., 2014).

See main text for additional details.

__________________________________________________________________ par(mfrow=c(1,2),cex=0.9) plot(eel.Blomberg_K,las=1,cex.axis=0.9) mtext("a)",adj=0,line=1) plot(eel.Pagel_lambda,bty="n",las=1,cex.axis=0.9,   xlim=c(0,1.1)) mtext("b)",adj=0,line=1)

Bayesian ancestral state estimation

The phytools package contains several different functions for discrete and continuous character ancestral state estimation under multiple models. Earlier, we reviewed the method of stochastic character mapping (Huelsenbeck, Nielsen & Bollback, 2003) and marginal ancestral character estimation, both of which are important tools for ancestral state reconstruction of discretely-valued traits.

Among the variety of approaches for ancestral character estimation of continuous characters that are implemented in the phytools package is the function anc.Bayes. As its name suggests, anc.Bayes performs ancestral state estimation using Bayesian MCMC. Just as any proper Bayesian approach should, the implementation of this method allows us to include prior information about the states at internal nodes. Here, I will illustrate the simplest type of analysis that we can undertake with the function in which I will simply accept the default node priors and MCMC conditions. anc.Bayes, however, will be most useful when we intend to explicitly incorporate prior knowledge about internal nodes of the tree—based on, for instance, observations from the fossil record.

To demonstrate the method, I will load a dataset (now packaged with phytools) that consists of a phylogeny and phenotypic trait information for a set of lizards from the family Cordylidae, originally published by Broeckhoven et al. (2016).

__________________________________________________________________ data(cordylid.tree) data(cordylid.data) head(cordylid.data)

##                    pPC1     pPC2     pPC3 ## C._aridus       0.59441 -0.40209  0.57109 ## C._minor        0.65171 -0.32732  0.55692 ## C._imkeae       0.19958 -0.08978  0.56671 ## C._mclachlani   0.62065  0.03746  0.86721 ## C._macropholis  0.44875 -0.75942  0.09737 ## C._cordylus    -0.07267  0.48294 -0.54394

Our trait data in this case are species scores for three different principal component (PC) axes from a phylogenetic principal components analysis undertaken using the phytools phyl.pca function (Revell, 2009). Cordylid lizards are known for their body and tail armor, consisting of large, rectangular scales called osteoderms. Principal component 1 in Broeckhoven et al. (2016) separated the most lightly armored cordylids (large negative values), from those cordylids with the heaviest body armor (large positive values of PC 1). Why don’t we extract this principal component from our data frame and rename it, as follows?

__________________________________________________________________ cordylid.armor score<-setNames(cordylid.data$pPC1,   rownames(cordylid.data))

With this named trait vector at the ready, we are prepared to undertake our Bayesian MCMC. As noted above, we will use the default conditions but update the number of generations that we want our MCMC to run to ngen=500000. Depending on the size of our phylogenetic tree, we may want to run more (or fewer) generations in a genuine empirical study.

__________________________________________________________________ cordylid.mcmc<-anc.Bayes(cordylid.tree,   cordylid.armor_score,ngen=500000)

## List of 7 ##  $ sig2  : num 0.713 ##  $ a     : num [1, 1] 0.000422 ##  $ y     : num [1:26] 0.000422 0.000422 0.000422 0.000422 ... ##  $ pr.mean: num [1:28] 1000 0 0 0 0 0 0 0 0 0 ... ##  $ pr.var: num [1:28] 1e+06 1e+03 1e+03 1e+03 1e+03 ... ##  $ prop  : num [1:28] 0.00713 0.00713 0.00713 0.00713 ... ##  $ sample: num 100 ## Starting MCMC... ## Done MCMC.

We can see that the method starts by printing out a summary of the “control parameters” of the MCMC. These include: initial values for the Brownian rate, σ2 (sig2), the root state (a), and the internal node values (y); information about our prior probability distributions (pr.mean and pr.var); the variances of the proposal distributions on each variable in the model (prop); and, finally, the interval that we will use to sample from our posterior distribution during the MCMC (sample). All of these parameters can be adjusted by the phytools user.

The object class that results from this function call (”anc.Bayes”) has a summary method in phytools that prints the mean from the posterior distribution, automatically excluding the first 20% of our samples as burn-in (though we can adjust this percentage if we would like). Though a thorough review of Bayesian MCMC is beyond the scope of this article, burn-in refers to the number of generations required for our MCMC to converge on the posterior probability distribution, and will depend on numerous factors including (but not limited to) our starting values, the parameter complexity of our model, and the proposal distribution. (See Roy, 2020 for a recent review of burn-in, convergence diagnostics, and related topics.) Convergence can be diagnosed quantitatively via multiple methods, including using the R package coda (Plummer et al., 2006). In addition to printing our results to screen, summary also passes the estimates (normally invisibly, but we can save them to a new variable in our workspace as we have done here) back to the user.

__________________________________________________________________ cordylid.ace<-summary(cordylid.mcmc)

## ## Object of class "anc.Bayes" consisting of a posterior ##    sample from a Bayesian ancestral state analysis: ## ## Mean ancestral states from posterior distribution: ##        29        30        31        32        33        34 ##  0.059277 -0.099027 -0.106452  0.057220  0.153671  0.201722 ##        35        36        37        38        39        40 ##  0.225081  0.297659  0.392992  0.493316  0.015503 -0.006053 ##        41        42        43        44        45        46 ##  0.435309  0.392526  0.300532  0.210391 -1.505181 -1.857682 ##        47        48        49        50        51        52 ## -0.136014 -0.520322 -0.829181 -0.985510 -1.040208  0.385293 ##        53        54        55 ##  0.511646  0.159943  0.028358 ## ## Based on a burn-in of 1e+05 generations.

Now that we have obtained our estimated Bayesian ancestral states for internal nodes, it is a straightforward task to visualize them on the branches and nodes of the tree. For this undertaking we will use the popular phytools plotting function contMap (Revell, 2013). By default, contMap uses Maximum Likelihood to compute ancestral states at all of the internal nodes of the tree—but it can also be supplied with user-specified values. Since we want to use our Bayesian estimates from anc.Bayes, that is what we will do here.

__________________________________________________________________ cordylid.contMap<-contMap(cordylid.tree,   cordylid.armor_score,anc.states=cordylid.ace,   plot=FALSE) cordylid.contMap<-setMap(cordylid.contMap,   viridisLite::viridis(n=10,direction=-1)) plot(cordylid.contMap,ftype="i",fsize=c(0.6,0.7),   leg.txt="PC 1 (increasing armor)",lwd=3) nodelabels(frame="circle",bg="white",cex=0.6)

For fun, compare Figs. 11 to 2 of Broeckhoven et al. (2016) in which estimated ancestral state values were assigned to each branch using a similar color gradient.

Figure 11 Reconstructed ancestral values from Bayesian MCMC projected onto the nodes and edges of the tree.

Numerical values at internal nodes are node indices from our input phylogeny. Data consist of PC 1 from a phylogenetic principal components analysis of cordylid morphological traits, and separate highly armored (high values) from lightly armored (low values) lizards (Broeckhoven et al., 2016). See main text for more details.

In addition to this simple analysis, we can (naturally) extract and plot posterior probability densities from any of our internal nodes of the tree. To see this, let us focus on the node labeled “49” in Fig. 11 and do exactly that. Node 49 corresponds to the common ancestor of the Pseudocordylus clade. The Pseudocordylus are among the most lightly armored of all cordylid lizards in this analysis, so we would expect our posterior distribution for this node to be centered on a relatively low value of our armor score.

__________________________________________________________________ cordylid.node49<-density(cordylid.mcmc,what=49) cordylid.node49

## ## Call: ##  density.anc.Bayes(x = cordylid.mcmc, what = 49) ## ## Data: node 49 (4001 obs.);   Bandwidth ’bw’ = 0.05679 ## ##        x                 y ##  Min.  :-2.1658   Min.  :0.000023 ##  1st Qu.:-1.5270   1st Qu.:0.022415 ##  Median:-0.8883   Median:0.187932 ##  Mean  :-0.8883   Mean  :0.391002 ##  3rd Qu.:-0.2495   3rd Qu.:0.784591 ##  Max.  : 0.3893   Max.  :1.187968

__________________________________________________________________ par(mar=c(5.1,4.1,1.1,2.1)) plot(cordylid.node49,las=1,bty="n",main="",cex.lab=0.8,   cex.axis=0.7,xlab="PC 1 (increasing armor)",   ylab="Posterior density",   xlim=range(cordylid.armor_score))

Figure 12 shows our estimate of the posterior probability distribution of the ancestral node 49 state, and should be centered precisely on the value we projected onto the tree of Fig. 11.

Figure 12 Posterior probability density at node 49 of Fig. 11 from Bayesian MCMC ancestral state reconstruction of PC 1 from a morphological analysis on a phylogenetic tree of cordylid lizards.

Node 49 corresponds to the common ancestor of the Pseudocordylus: a relatively lightly armored cordylid clade. See main text for more details.

As given here, Bayesian MCMC ancestral state reconstruction will yield (in nearly all circumstances) point estimates that are highly similar to the values that we might have obtained using Maximum Likelihood. Nonetheless, Bayesian inference provides the additional benefit of supplying a natural framework for incorporating prior information about the states at one or various internal nodes in the tree (by adjusting pr.mean, pr.var, or both: see above), as well as for measuring the substantial uncertainty that can be associated with ancestral trait estimates (in particular, by providing not just confidence intervals around each node, but sets of ancestral values across all nodes of the tree that have been sampled in proportion to their posterior probability under the model).

Multivariate trait evolution

Along with the various univariate methods we have seen so far, phytools also contains a handful of different multivariate trait evolution models, designed for both continuous and discrete characters.

One of these is an interesting model (described in Revell & Collar, 2009; Revell, Toyama & Mahler, 2022) in which the rates and evolutionary correlations between traits are allowed to vary as a function of a set of mapped regimes on the tree. (Similar to O’Meara et al., 2006, but for more than one trait at a time.) The underlying motivation of this method is to test hypotheses about phylogenetic heterogeneity in the evolutionary relationship (i.e., correlation) between different traits on our phylogeny. This approach is also used to study quantitative trait modularity and integration during macroevolution (e.g., Damian-Serrano, Haddock & Dunn, 2021).

Note that closely related analyses have been implemented in the R packages mvMORPH by Clavel, Escarguel & Merceron (2015), and ratematrix by Caetano & Harmon (2017). The packages mvSLOUCH (Bartoszek et al., 2012), PhylogeneticEM (Bastide et al., 2018), and PCMFit (Mitov, Bartoszek & Stadler, 2019) also feature phylogenetic multivariate quantitative trait analysis methods.

To illustrate our approach, however, I will use a phylogenetic tree and dataset of tropidurid lizard species from Revell, Toyama & Mahler (2022).

__________________________________________________________________ data(tropidurid.tree) data(tropidurid.data)

In this case, our phylogeny is already a tree with mapped regimes. We can see this by merely printing the model object that we loaded. (In an empirical study we might imagine using a set of such trees sampled in proportion to their probabilities using stochastic mapping—and then averaging the result e.g., see Revell & Harmon, 2022.)

__________________________________________________________________ print(tropidurid.tree,printlen=2)

## ## Phylogenetic tree with 76 tips and 75 internal nodes. ## ## Tip labels: ##  Leiocephalus_raviceps, Leiocephalus_carinatus, ... ## ## The tree includes a mapped, 2-state discrete character ## with states: ##  n_rock, rock ## ## Rooted; includes branch lengths.

This tells us that our phylogenetic tree contains 76 taxa and a mapped regime with two states: ”n_rock” (non-rock dwelling) and “rock” (rock-dwelling). Since phytools permits mapped regimes to have arbitrarily lengthy names, let us rename these two regime levels in a more informative way. To do so, I will use the phytools function mergeMappedStates. mergeMappedStates, as readers can probably guess, is designed to merge the mappings of two or more traits into one –but can also be employed to simply substitute one mapping name for another.

__________________________________________________________________ tropidurid.tree<-mergeMappedStates(tropidurid.tree,   "n_rock","non-rock dwelling") tropidurid.tree<-mergeMappedStates(tropidurid.tree,   "rock","rock-dwelling")

Let us plot this updated tree. To do so, I am going to use the recent phytools function sigmoidPhylogram that will plot our tree using curved (“sigmoidal”) linking lines (Fig. 13). phytools contains lots of cool tree plotting functions like this one.

Figure 13 Phylogenetic tree of rock- and non-rock dwelling tropidurid lizard species from Revell, Toyama & Mahler (2022).

Mapped colors correspond to a hypothesis of the history of habitat use across the clade. See main text for more details.

__________________________________________________________________ cols<-setNames(c("white","black"),c("non-rock dwelling",   "rock-dwelling")) sigmoidPhylogram(tropidurid.tree,direction="upwards",   outline=TRUE,colors=cols,direction="upwards",   outline=TRUE,lwd=2,fsize=0.4,ftype="i",offset=1) legend("bottomright",c("non-rock dwelling",   "rock-dwelling"),pch=22,pt.bg=cols,cex=0.8,   pt.cex=1.2)

Our quantitative phenotypic trait data in tropidurid.data consist of a single measure of overall body size (as trait 1, “newsize”), and a second metric trait measuring dorsoventral depth (vs. flattening, ”body_height”).

__________________________________________________________________ head(tropidurid.data)

##                            newsize body_height ## Leiocephalus_raviceps     2.358317  0.05768818 ## Leiocephalus_carinatus    2.931721  0.30216220 ## Leiocephalus_psammodromus 2.700397  0.19667083 ## Leiocephalus_personatus   2.535315  0.32216983 ## Leiocephalus_barahonensis 2.473666  0.30266917 ## Stenocercus_ochoai        2.549010  0.29067644

Our hypothesis of multivariate trait evolution in this clade is that these two traits (size and body depth) should generally scale together in non-rock dwelling lizard species: bigger lizards also tend to have larger body depths. We hypothesize, however, that this general relationship may become decoupled in rock-dwelling lineages where the force of selection is predicted to favor increased flattening, relative to their non-rock dwelling kin. (There are biomechanical and behavioral reasons to suspect this could be so. For more information, see Revell et al., 2007; Revell, Toyama & Mahler, 2022.)

To test this hypothesis, we will use the phytools function evolvcv.lite which fits a hierarchical set of models for the evolutionary rates (of each character) and evolutionary correlations (between them, Revell & Collar, 2009; Revell, Toyama & Mahler, 2022). Following Revell & Collar (2009), these models are: (1) a model with common rates and correlations between the two discrete traits; (2) a model with different rates of evolution depending on our mapped state, but a common correlation; (3) a model with common rates, but a different evolutionary correlation, depending on the mapped discrete character; and, finally, (4) a model of different rates and correlations between the two discrete mapped character states.

__________________________________________________________________ tropidurid.fits<-evolvcv.lite(tropidurid.tree,   tropidurid.data)

## Fitting model 1: common rates, common correlation... ## Best log(L) from model 1: 52.3056. ## Fitting model 2: different rates, common correlation... ## Best log(L) from model 2: 54.3968. ## Fitting model 3: common rates, different correlation... ## Best log(L) from model 3: 55.1105. ## Fitting model 4: no common structure... ## Best log(L) from model 4: 56.2877.

Having fit each of four models (in this case: evolvcv.lite actually includes several additional models that we will not review here, see Revell, Toyama & Mahler, 2022 for more details), we can most easily compare all of the models in our set using a generic anova function call as follows.

__________________________________________________________________ anova(tropidurid.fits)

##           log(L) d.f.       AIC     weight ## model 1 52.30560    5 -94.61119 0.09221085 ## model 2 54.39681    7 -94.79362 0.10101737 ## model 3 55.11048    6 -98.22096 0.56057434 ## model 4 56.28765    8 -96.57530 0.24619744

This comparison shows us that our third model (”model 3”: remember, with common rates but different evolutionary correlation between rock and non-rock dwelling species) is the best-supported explanation of our data in this set, with the lowest AIC score and highest model weight. We can print out a summary of our set of four models to review the estimated parameter values of each.

__________________________________________________________________ tropidurid.fits

## Model 1: common rates, common correlation ##  R[1,1]  R[1,2]  R[2,2]  k   log(L)  AIC ## fitted   0.2224  0.0154  0.0589  5   52.3056 -94.6112 ## ## (R thinks it has found the ML solution for model 1.) ## ## Model 2: different rates, common correlation ##  R[1,1]  R[1,2]  R[2,2]  k   log(L)  AIC ## non-rock dwelling    0.2025  0.0187  0.0456  7   54.3968 -94.7936 ## rock-dwelling    0.3043  0.0382  0.1263 ## ## (R thinks it has found the ML solution for model 2.) ## ## Model 3: common rates, different correlation ##  R[1,1]  R[1,2]  R[2,2]  k   log(L)  AIC ## non-rock dwelling    0.2256  0.0394  0.0588  6   55.1105 -98.221 ## rock-dwelling    0.2256  -0.0354 0.0588 ## ## (R thinks it has found the ML solution for model 3.) ## ## Model 4: no common structure ##  R[1,1]  R[1,2]  R[2,2]  k   log(L)  AIC ## non-rock dwelling    0.2108  0.0325  0.0485  8   56.2877 -96.5753 ## rock-dwelling    0.2794  -0.0564 0.101 ## ## (R thinks it has found the ML solution for model 4.)

Here we see that model 3 is one in which the evolutionary covariance between overall body size and dorsoventral flattening is negative among rock-dwelling lineages –compared to the positive evolutionary covariance in non-rock species and across all other models. Just as we had predicted, size and body depth are evolutionarily decoupled in rock-dwelling specialists.

Variable rate Brownian motion

Lastly, I recently added a function to phytools that permits us to fit a variable-rate Brownian evolution model using penalized likelihood (Revell, 2021). Related methods have been implemented both outside (e.g., Venditti, Meade & Pagel, 2011) and inside (e.g., Uyeda & Harmon, 2014; Martin et al., 2022) R.

In our model, we will assume that the phenotypic trait evolves via a standard Brownian motion process—but that the rate of evolution (σ2) itself also changes through time and among the clades of our tree via a process of geometric Brownian motion. (That is, Brownian motion on a log scale.) As one might expect for a penalized likelihood method, when we go ahead and fit this model to data, the degree to which the evolutionary rate is permitted to vary from edge to edge in the tree is controlled by our λ penalty or “smoothing” coefficient (Revell, 2021).

Although a relatively new addition to the phytools package, this method has already been used to, for example, investigate rate heterogeneity differences in body size evolution between cetaceans and plesiosaurs (Sander et al., 2021), and to measure rate variation in the evolution of the mechanical properties of woody plant tissue (Higham, Schmitz & Niklas, 2022). Here, I will apply it to the analysis of skull size evolution in a phylogenetic tree of primates. My data for this example (now packaged with phytools) come from a book chapter authored by Kirk & Kay (2004).

__________________________________________________________________ data(primate.tree) data(primate.data)

Our data frame, primate.data, contains a number of different variables. Let us pull out just one of these, Skull_length, and (as we do) convert it to a logarithmic scale.

__________________________________________________________________ primate.lnSkull<-setNames(   log(primate.data$Skull_length),   rownames(primate.data)) head(primate.lnSkull)

## Allenopithecus_nigroviridis           Alouatta_palliata ##                    4.590057                    4.698661 ##          Alouatta_seniculus           Aotus_trivirgatus ##                    4.682131                    4.102643 ##           Arctocebus_aureus     Arctocebus_calabarensis ##                    3.901973                    3.985273

With just this input data vector and our tree, we are already ready to run our penalized likelihood analysis. As I mentioned earlier, however, penalized likelihood requires the user to specify a smoothing parameter—normally denominated λ. λ determines the weight that is assigned to the penalty term of the fitted model, in our case a measure of how much (or how little) the evolutionary rate evolves from edge to edge in the phylogeny (Revell, 2021). A large value of λ will more stringently penalize high rate variation between edges and thus cause us to fit a model with relatively low rate heterogeneity across the tree. Smaller values of λ, on the other hand, should have the converse effect.

A number of approaches, such as cross-validation (e.g., Efron & Gong, 1983), have been suggested to help us identify suitable values of λ in penalized likelihood for our data and question—however, I’d minimally recommend testing multiple values of λ and comparing the results. Let us do exactly that for our analysis of primate skull length: first using λ = 1.0, and then swapping it for a much smaller λ = 0.1 and much larger λ = 10. This will allow us to pretty quickly see how these different values of our smoothing parameter affect our findings, and thus how sensitive any inference we draw might be to the specific value of λ we assigned.

Before continuing, however, we will try to get a better sense of our data by creating a simple projection of our phenotypic trait (log skull length) onto the tree. Visual inspection may help give us a preliminary sense of where in our tree our penalized likelihood method could end up showing the rate of primate skull length evolution to vary the most –and the least. In this case, I will use two different plotting methods.

First, I will use the phytools function edge.widthMap which sizes the thickness of our plotted branches in proportion to the observed or reconstructed trait values. (This is one of my favorite phytools functions, but, compared to the contMap method we saw earlier, so far as I can tell it has been used very little in published literature.) We can see the result in Fig. 14A. In addition to this visualization, I will also undertake a simple projection of our phylogeny into the trait space. This is done using a very popular phytools plotting method called phenogram (Evans et al., 2009; Revell, 2013; Revell, 2014b). In the typical style of this kind of plot, our phylogeny is graphed in a space defined by time since the root of the tree (on our horizontal axis), and the observed or reconstructed values of our phenotypic trait (on the vertical, Revell, 2013). The result of this projection is shown in Fig. 14B.

Figure 14 (A) Primate skull lengths (on a log scale) projected onto the edges and nodes of the phylogeny. The width of each edge of the tree is proportional to the observed or reconstructed value of the trait. See main text for more details. (B) A projection of the phylogeny of (A) into a space defined by time since the root (on the horizontal axis) and log skull length.

Phylogeny and data are derived from Kirk & Kay (2004). See main text for more details.

__________________________________________________________________ par(mfrow=c(1,2)) primate.widthMap<-edge.widthMap(primate.tree,   primate.lnSkull) plot(primate.widthMap,color=palette()[4],   legend="log(skull length)",border=TRUE,fsize=0.4,   mar=c(4.1,1.1,2.1,0.1)) mtext("a)",adj=0,line=0,cex=1.4) phenogram(primate.tree,primate.lnSkull,fsize=0.4,   ftype="i",spread.cost=c(1,0),mar=c(4.1,4.1,2.1,0.1),   quiet=TRUE,las=1,cex.axis=0.8,   ylab="log(skull length)") mtext("b)",adj=0,line=0,cex=1.4)

The function we will use to fit our rate-variable model, multirateBM, performs a computationally intensive optimization. Setting the optional argument parallel=TRUE will help distribute this burden across multiple processors of our computer, if possible. Let us start our analysis using a smoothing parameter, λ, equal to λ = 1.0.

__________________________________________________________________ primate.mBM 1<-multirateBM(primate.tree,   primate.lnSkull,lambda=1,parallel=TRUE)

## Beginning optimization.... ## Using socket cluster with 16 nodes on host ’localhost’. ## Optimization iteration 1. Using "L-BFGS-B" (parallel) ## optimization method. ## Best (penalized) log-likelihood so far: -267.108 ## Done optimization.

Now we can do the same with λ = 0.1 and 10. Readers should take special care to note that the specific values of the penalized log likelihoods are not comparable between analyses with different values of the penalty coefficient, λ. This time I will turn off printing by updating the optional argument to quiet=TRUE.

__________________________________________________________________ primate.mBM 0.1<-multirateBM(primate.tree,   primate.lnSkull,lambda=0.1,parallel=TRUE,quiet=TRUE) primate.mBM_10<-multirateBM(primate.tree,   primate.lnSkull,lambda=10,parallel=TRUE,quiet=TRUE)

Finally, let us visualize the differences and similarities between each of our three fitted models.

__________________________________________________________________ par(mfrow=c(1,3)) plot(primate.mBM_1,ftype="off",lwd=2,   mar=c(0.1,0.1,2.1,0.1)) mtext(expression(paste("a) ",lambda," = 1")),adj=0.1,   line=0.5,cex=1.1) plot(primate.mBM_0.1,ftype="off",lwd=2,   mar=c(0.1,1.1,2.1,0.1)) mtext(expression(paste("b) ",lambda," = 0.1")),adj=0.1,   line=0.5,cex=1.1) plot(primate.mBM_10,ftype="off",lwd=2,   mar=c(0.1,1.1,2.1,0.1)) mtext(expression(paste("c) ",lambda," = 10")),adj=0.1,   line=0.5,cex=1.1)

We can see from the plot of Fig. 15 that even though the specific range of rate variation depends strongly on our specified values of λ, the pattern from clade to clade on the tree is relatively robust. This should give us some measure of confidence that the our inferred rate heterogeneity may be a product of real variability in the evolutionary rate for our character on the phylogeny.

Figure 15 Estimated rates of log (skull length) evolution in primates under a variable-rate Brownian evolution model for different values of the smoothing parameter, λ.

Increasing values of λ should correspond to less variation in the rate of evolution across the tree. Phylogeny and data are based on Kirk & Kay (2004). See main text for additional details.

Diversification

In addition to the methods that we have seen so far, phytools also contains a handful of different techniques for investigating diversification on reconstructed phylogenies. Diversification has never been the primary focus of the phytools R package (to that end, I would recommend the powerful diversitree package, FitzJohn, 2012), but these methods are popular, and the phytools implementations can be relatively easy to use. Various additional R packages include interesting diversification models and methods, such as hisse (Beaulieu & O’Meara, 2016), RPANDA (Morlon et al., 2016), TreeSim (Stadler, 2019), DDD (Etienne & Haegeman, 2023), and others.

phytools contains methods to compute and visualize the accumulation of lineages through time, including with extinction (ltt), to calculate and test the γ statistic (gammatest, mccr, Pybus & Harvey, 2000), to fit pure-birth and birth-death models, including with random missing taxa (fit.yule and fit.bd, Nee, May & Harvey, 1994; Stadler, 2013), to compare diversification rates between trees (ratebytree, Revell, 2018), and to simulate stochastic trees under various conditions (pbtree).

Lineage through time plots

One of the most rudimentary phylogenetic methods for studying diversification is to simply graph the accumulation of new lineages in our reconstructed phylogeny over time since the global root of the tree. This visualization method is called a lineage-through-time plot. A great appeal of this visualization is that if we graph the number of lineages through time in a fully-sampled pure-birth (that is, constant-rate speciation, but no extinction) phylogenetic tree, the accumulation curve should be exponential –or exactly linear on a semi-logarithmic scale. This means that the lineage-through-time plot gives us a handy tool that we can use to compare the real lineage accumulation in our reconstructed tree to this simple, neutral expectation (Pybus & Harvey, 2000; Revell & Harmon, 2022).

To see how the number of lineages through time are calculated and graphed using phytools, let us load a phylogenetic tree of snakes from the venomous family Elapidae. This phylogeny is now packaged with phytools but derives from a study by Lee et al. (2016).

__________________________________________________________________ data(elapidae.tree) print(elapidae.tree,printlen=2)

## ## Phylogenetic tree with 175 tips and 174 internal nodes. ## ## Tip labels: ##   Calliophis_bivirgata, Calliophis_melanurus, ... ## ## Rooted; includes branch lengths.

We are going to create our lineage-through-time graph with phytools over two steps. First, we will use the phytools function ltt to compute an object of class “ltt” containing our tree and a count of the number of lineages through time from the root of the tree to the tips.

__________________________________________________________________ elapidae.ltt<-ltt(elapidae.tree,plot=FALSE) elapidae.ltt

## Object of class "ltt" containing: ## ## (1) A phylogenetic tree with 175 tips and 174 internal ##     nodes. ## ## (2) Vectors containing the number of lineages (ltt) and ##     branching times (times) on the tree. ## ## (3) A value for Pybus & Harvey’s "gamma" statistic of ##     gamma = -3.3244, p-value = 9e-04.

From the print-out we see that in addition to the tree and the lineages through time, our object also contains a value of (and a P-value for) Pybus & Harvey’s (2000) Pybus & Harvey’s (2000) γ statistic. γ is a numerical value used to describe the general shape of the lineage through time curve. If the curve is straight (on a semi-log scale), then γ should have a value close to zero. This is what we expect under a pure-birth (speciation only) diversification process. On the other hand, significantly positive or significantly negative γ mean that the lineage through time graph curves upward or downward towards the present day (Pybus & Harvey, 2000). Significant positive or negative curvature of the lineage-through-time plot might mean that the rate of diversification has changed over time, but it could also be due to past extinction or incomplete taxon sampling (Revell & Harmon, 2022). Note that since the pull of the present (Nee, Mooers & Harvey, 1992) means that our lineage through time plot is expected to curve upwards towards the present day for any non-zero rate of extinction, some have argued that γ should only be interpreted when negative (i.e., that statistical tests of γ are properly one-tailed). I do not subscribe to that view, inasmuch as I see γ as a phenomenological measure of lineage accumulation in our reconstructed tree whose positive or negative deviation from the statistic’s expected value under pure-birth could have multiple underlying causes. At first look, the value of γ from our elapid snake phylogeny would seem to be highly significantly negative.

To proceed and graph our object created in the previous step, we merely need to execute a generic plot method function call as follows. A simple plot call would have done the trick; however, in this case I decided to first leave off the axes of my plot, and then re-plot them so that I could make our horizontal (x) axis run backwards in time (i.e., right to left) from the present day into the past. I have also super-imposed the phylogeny itself on our plot so that we can more easily visualize the relationship between the structure of our phylogenetic tree and the accumulation of lineages over time.

__________________________________________________________________ par(mar=c(5.1,4.1,1.1,2.1)) plot(elapidae.ltt,show.tree=TRUE,lwd=2,   log.lineages=FALSE,log="y",bty="n",cex.lab=0.9,   transparency=0.1,axes=FALSE,   xlab="millions of year bp") h<-max(nodeHeights(elapidae.tree)) axis(1,at=h-seq(0,35,by=5),labels=seq(0,35,by=5),las=1,   cex.axis=0.8) axis(2,las=1,cex.axis=0.8)

In general, accounting for incomplete taxon sampling in the measurement of the γ statistic is important because missing taxa will tend to pull our lineage-through-time curve downwards as we approach the tips of the tree –in other words, towards more negative values of γ, just like the value that we see for our lineage through time plot of Fig. 16.

Figure 16 Lineage through time plot for phylogeny of snakes from the family Elapidae (Lee et al., 2016).

See main text for more details.

Fortunately, there is a simple way to address this bias. If we know the true species richness of our clade of interest, we can simply simulate trees that match this richness under pure-birth, randomly prune taxa to the level of “missingness” in our reconstructed tree, and then use the distribution of γ values across this set of simulated (and then randomly pruned) trees as our null distribution for hypothesis testing. This exact procedure is called the “Monte Carlo constant rates” (MCCR, Pybus & Harvey, 2000) test and is implemented in the phytools function mccr.

Of course, since the MCCR test accounts for randomly missing taxa from our tree, we must know or hypothesize a true species richness of our clade. In this instance, we are not too preoccupied about the precise value for Elapidae, but Lee et al. (2016) purported that their phylogeny included approximately 50% of known elapids at the time. Even though it is likely that elapid diversity has changed a bit in the intervening years, for illustrative purposes only, let us just go with this 50% figure. In both mccr and the birth-death model-fitting function we will use later, sampling fraction is specified via the argument rho (for the Greek letter ρ).

__________________________________________________________________ elapidae.mccr<-mccr(elapidae.ltt,rho=0.5,nsim=1000) elapidae.mccr

## Object of class "mccr" consisting of: ## ## (1) A value for Pybus & Harvey’s "gamma" statistic of ##     gamma = -3.3244. ## ## (2) A two-tailed p-value from the MCCR test of 0.446. ## ## (3) A simulated null-distribution of gamma from 1000 ##     simulations.

This tells us that, having accounted for missing taxa, our observed value of γ (previously highly significantly negative) becomes indistinguishable from what we would expect under pure-birth. We can plot our results to see what I mean.

__________________________________________________________________ par(mar=c(5.1,4.1,0.6,2.1)) plot(elapidae.mccr,las=1,cex.lab=0.8,cex.axis=0.7,   main="")

Figure 17 shows that the measured value of γ by the MCCR test is no longer significant, demonstrating the vital importance of accounting for incomplete taxon sampling in this (and other) diversification analyses using phylogenies.

Figure 17 Distribution of simulated values of γ for the MCCR test, and observed value for the lineage through time curve of the phylogeny of elapid snakes given in Fig. 16.

Phylogenetic tree based on Lee et al. (2016). See main text for more details.

Modeling speciation and extinction

In addition to these analysis, phytools can also fit simple speciation and extinction models following Nee, May & Harvey (1994), Stadler (2013), Harmon (2019). This is done primarily using the function fit.bd, which also allows us to take into account an incomplete taxonomic sampling fraction (Stadler, 2013).

Just as with γ, incomplete sampling has the potentially to substantially distort our estimated rates of speciation (normally given as λ—a different λ from before) and extinction (μ). In this case, ignoring (or underestimating) the missing lineages in our tree will tend to cause us to underestimate the rate of extinction, as nearly all of the information we have about extinction comes from the most recent parts of our phylogeny. (See Stadler, 2013; Harmon, 2019; Revell & Harmon, 2022 for more details.)

Fitting a birth-death model using phytools is very easy. For this example, we will use phylogenetic tree of lizards from the diverse South American family Liolaemidae. Just as in the other examples this phylogeny is packaged with phytools, but was originally published by Esquerré et al. (2019). (This is the same phylogeny that was used to study parity mode evolution under the hidden rates model in an earlier section.)

__________________________________________________________________ data(liolaemid.tree) print(liolaemid.tree,printlen=2)

## ## Phylogenetic tree with 257 tips and 256 internal nodes. ## ## Tip labels: ##   Liolaemus_abaucan, Liolaemus_koslowskyi, ... ## ## Rooted; includes branch lengths.

We will pass our liolaemid tree to the fit.bd function, and the only additional argument to be assigned is rho (for ρ), the sampling fraction, just as we did for the MCCR test in the function mccr. The Reptile Database (Uetz et al., 2023) puts the total species richness of Liolaemidae at 341, so we can set rho to have a value equal to the number of tips in our tree divided by this quantity.

__________________________________________________________________ liolaemid.rho<-Ntip(liolaemid.tree)/341 liolaemid.bd<-fit.bd(liolaemid.tree,rho=liolaemid.rho) liolaemid.bd

## ## Fitted birth-death model: ## ## ML(b/lambda) = 0.352 ## ML(d/mu) = 0.1781 ## log(L) = 526.451 ## ## Assumed sampling fraction (rho) = 0.7537 ## ## R thinks it has converged.

Other R packages (such as the aforementioned diversitree) might allow us to compare our fitted birth-death model to a range of other hypotheses about diversification, such as that the speciation and extinction rates change through time or as a function of our phenotypic traits (e.g., Maddison, Midford & Otto, 2007; FitzJohn, 2010; Morlon, Potts & Plotkin, 2010; Revell & Harmon, 2022). In phytools we can compare our fitted birth-death model to only one alternative model: the simpler, pure-birth model—also called a ‘Yule’ model.

__________________________________________________________________ liolaemid.yule<-fit.yule(liolaemid.tree,   rho=liolaemid.rho) liolaemid.yule

## ## Fitted Yule model: ## ## ML(b/lambda) = 0.2502 ## log(L) = 521.1832 ## ## Assumed sampling fraction (rho) = 0.7537 ## ## R thinks it has converged.

__________________________________________________________________ anova(liolaemid.yule,liolaemid.bd)

##                  log(L) d.f.       AIC     weight ## liolaemid.yule 521.1832    1 -1040.366 0.01381943 ## liolaemid.bd   526.4510    2 -1048.902 0.98618057

This result tells us that, in the context of the two very simple models that we have fit to our reconstructed tree, a two-parameter birth-death (speciation and extinction) model is much better supported than our simpler Yule model.

Lastly, the phytools function fit.bd exports a likelihood function as part of the fitted model object. This, in turn, makes it very straightforward for phytools users to (for example) compute and graph the likelihood surface. Here, I will illustrate this using the base R graphics function persp. (But R and contributed R packages contain lots of even fancier 3D plotting methods that readers might be more interested in trying.)

__________________________________________________________________ ngrid<-40 b<-seq(0.25,0.45,length.out=ngrid) d<-seq(0.10,0.25,length.out=ngrid) logL<-matrix(NA,ngrid,ngrid) for(i in 1:ngrid) for(j in 1:ngrid)   logL[i,j]<-liolaemid.bd$lik(c(b[i],d[j])) logL[is.nan(logL)]<-min(logL[!is.nan(logL)]) par(mar=rep(0.1,4)) persp(b,d,exp(logL),shade=0.3,phi=45,theta=20,   xlab="speciation rate",ylab="extinction rate",   zlab="likelihood",border=palette()[4],expand=0.3)

Some astute readers will notice the line logL[is.nan(logL)] <- min(...) (etc.) in my script of above. This is because during our grid evaluation of the likelihood function, sometimes the function was being evaluated in parameter space where the likelihood is not defined. To account for this I set all parts of the likelihood surface that could not be computed to the numerical minimum of the graph.

Figure 18 shows the very strong ridge in the likelihood surface (from low λ and low μ, to high λ and high μ) that almost invariably tends to characterize the likelihood surfaces of birth-death models.

Figure 18 Visualization of the likelihood surface for speciation and extinction rates estimated for a phylogenetic tree of Liolaemidae.

The ridge of values with similar likelihoods is typical of this class of model. Phylogenetic tree from Esquerré et al. (2019). See main text for more details.

Visualization

After phylogenetic comparative analysis, phytools is perhaps best known for its phylogeny visualization methods, and we have seen a number of these approaches already deployed throughout this article. For example, in Figs. 1, 2, 3, 4, 7 and 13 I illustrated custom phytools plotting methods for stochastic character mapping and the analysis of stochastically mapped trees. Likewise, in Figs. 5, 6 and 9 I demonstrated phytools plotting methods for fitted discrete character evolution models. In Figs. 10, 11, 14 and 15 I showed a variety of custom methods for visualizing continuous trait evolution. Finally, in Figs. 8, 16 and 17 I illustrated several different approaches for graphing diversification or the results from an analysis of diversification on the tree. This is a sparse sample of the variety of plotting methods for phylogenies, phylogenetic comparative data, and the results of phylogenetic analysis that are implemented in the phytools package.

In this final section, I will illustrate just a few more popular plotting methods of the package that we have not already seen in prior bits of the present article.

Co-phylogenetic plotting

Among the most popular plotting method of the phytools package is the function cophylo, which creates co-phylogenetic plots (often referred to as “tanglegrams,” Page, 1993).

The purpose of tanglegrams varies widely from study to study. Classically, for instance, tanglegrams have been used to visually illustrate the topological similarity between two groups that are hypothesized to co-speciate: for instance, an animal host and its parasites, or a plant and its pollinators (e.g., Page, 1993; Medina & Langmore, 2016; Endara et al., 2018; Caraballo, 2022).

Equally often, however, tanglegrams are put to different purposes. For instance, tanglegrams are frequently employed to show the similarity or differences between alternative phylogenetic hypotheses (e.g., Amarasinghe et al., 2021), to identify incongruence among gene trees (e.g., Stull et al., 2020), and even to compare a phylogenetic history to a non-phylogenetic cluster dendogram based on phenotypic or ecological data (e.g., Atkinson, Ee & Pfeiffer, 2020; Huie et al., 2021). To illustrate the phytools tanglegram method, I will use a phylogenetic tree of bat species and another of their betacoronaviruses—both based on Caraballo (2022).

__________________________________________________________________ data(bat.tree) data(betaCoV.tree)

Assuming that our tip labels differ between our different trees (and they do in this instance), we need more than just two phylogenies to create a tanglegram–we also need a table of associations linking the tip labels of one tree to those of the other. Again, based on Caraballo (2022), our association information for the two trees that we have loaded is contained in the phytools data object bat_virus.data. Let us load and review it.

__________________________________________________________________ data(bat virus.data) head(bat_virus.data)

##                    Bats betaCoVs ## 1    Artibeus lituratus KT717381 ## 2 Artibeus planirostris MN872692 ## 3 Artibeus planirostris MN872690 ## 4 Artibeus planirostris MN872691 ## 5 Artibeus planirostris MN872689 ## 6 Artibeus planirostris MN872688

Inspecting just the first part of this object reveals its general structure. We can see that it consists of two columns: one for each of our two trees. The elements of the first column should match the labels of our first tree, and those of the second column the labels of our second tree. There is no problem at all if one or the other column has repeating names: a host can (of course) be associated with more than one parasite, and vice versa.

Now let us run our co-phylogenetic analysis. This will create, not a plot, but a “cophylo” object in which the node rotation has been optimized to maximize the tip alignment of the two trees.

__________________________________________________________________ bat.cophylo<-cophylo(bat.tree,betaCoV.tree,   assoc=bat_virus.data)

## Rotating nodes to optimize matching... ## Done.

We can print this object, as follows.

__________________________________________________________________ bat.cophylo

## Object of class "cophylo" containing: ## ## (1) 2 (possibly rotated) phylogenetic trees in an object of class ##     "multiPhylo". ## ## (2) A table of associations between the tips of both trees.

To plot it, we will use the a generic phytools plot method for the object class. I will go ahead and adjust a few settings of the method to make our graph look nice—and I will use species-specific linking line colors so that we can more easily visualize all the different virus sequences that are associated with each bat host. (My color palette comes from the RColorBrewer function brewer.pal Neuwirth, 2022. I chose to use RColorBrewer here, rather than the viridis palette from earlier in the article, because it creates aesthetic divergent color palettes—whereas viridis will create a color gradient. RColorBrewer can be installed from CRAN in the typical way.)

__________________________________________________________________ cols<-setNames(RColorBrewer::brewer.pal(n=7,   name="Dark2"),bat.tree$tip.label) par(lend=3) plot(bat.cophylo,link.type="curved",fsize=c(0.7,0.6),   link.lwd=2,link.lty="solid",pts=FALSE,   link.col=make.transparent(cols[bat_virus.data[,1]],     0.5),ftype=c("i","reg")) pies<-diag(1,Ntip(bat.tree)) colnames(pies)<-rownames(pies)<-names(cols) tiplabels.cophylo(pie=pies,   piecol=cols[bat.cophylo$trees[[1]]$tip.label],   which="left",cex=0.2)

In general, our plot of Fig. 19 reveals a surprisingly strong association between the topology of the phylogenies of the bats and their viruses—a pattern that Caraballo (2022) also reported (and that happened to contrast with what Caraballo found for alphacoronaviruses, for what it is worth).

Figure 19 Co-phylogenetic plot of bat species (left) and their associated betacoronaviruses (right, labeled by GenBank accession number).

Associations and GenBank accession numbers from Caraballo (2022). See main text for more details.

Projecting a tree onto a geographic map

phytools can also be used to project a phylogenetic tree onto a geographic map, a visualization technique that has been used in numerous published studies since it was added to the package (e.g., Csosz & Fisher, 2016; Quach, Reynolds & Revell, 2019; Hermanson et al., 2020; Huang & Morgan, 2021; Osuna-Mascaró et al., 2023).

To see how this is done in R, we will load two datasets that come with the phytools package. The first is a phylogenetic tree of Galapagos giant tortoises (genus Chelonoidis, tortoise.tree) based on nucleotide sequence data published Poulakakis et al. (2020). The second is a corresponding geographic dataset that I obtained from Fig. 1 of the same study (Poulakakis et al., 2020).

__________________________________________________________________ data(tortoise.tree) data(tortoise.geog)

Our geographic data (tortoise.geog), which contain latitude and longitude measures in two columns, can be a data frame or matrix. In the event that any of the operational taxa of our tree are represented more than once in our geographic data, then our coordinate data must take the form of a matrix. This is important to note because our plotting function requires that our taxon labels be supplied as row names. The most common ways to read data into R (for instance, using read.table or read.csv) create data frames, rather than a matrices—and R data frames do not permit repeating row names. In the case of our tortoise data, the labels of our data and tree match without duplication, so our input data can be provided in either acceptable format.

Let us review our locality data frame, tortoise.geog, to understand precisely how it has been structured.

__________________________________________________________________ tortoise.geog

##                        lat      long ## C._duncanensis_1 -0.611014 -90.66008 ## C._abingdonii     0.583058 -90.75376 ## C._niger         -1.291984 -90.42749 ## C._vicina_1      -0.915375 -91.38897 ## C._chathamensis  -0.818184 -89.41856 ## C._becki          0.031506 -91.39121 ## C._darwini       -0.268896 -90.70471 ## C._donfaustoi    -0.642738 -90.20561 ## C._hoodensis     -1.378876 -89.67889 ## C._duncanensis_2 -0.611014 -90.66008 ## C._porteri       -0.697202 -90.48687 ## C._vicina_2      -0.915375 -91.38897 ## C._guntheri      -0.800044 -91.03839 ## C._vanderburghi  -0.447016 -91.10362 ## C._microphyes    -0.250360 -91.32202

We should see that it consists of species names as row names, as promised, and geographic locality points in the form of decimal latitude (in the first column) and longitude (in the second) coordinates.

Our next step will be to build the map projection that we intend to plot. This is done using the phytools function phylo.to.map. In addition to combining our phylogenetic tree and map data, phylo.to.map, much like the cophylo method of the previous section, performs a series of node rotations designed to optimize the alignment of our phylogeny with the geographic coordinates of our tip data. As node rotation is arbitrary anyway, this can be helpful to facilitate a more convenient visualization.

Before running this code section, we will load the R package mapdata (Becker, Wilks & Brownrigg, 2022b), which can be installed from CRAN in the usual way). This will allow us to access a higher resolution base map of the geographic region we intend to plot. We should also specify direction=“rightwards”. This indicates that we intend to graph our phylogeny to the left of our plotted map facing right, and thus permits phylo.to.map to optimize its node rotations of the tree accordingly.

__________________________________________________________________ library(mapdata) tortoise.phymap<-phylo.to.map(tortoise.tree,   tortoise.geog,plot=FALSE,direction="rightwards",   database="worldHires",regions="Ecuador")

## objective: 64

## objective: 52 ## objective: 52 ## objective: 52 ## objective: 52 ## objective: 52 ## objective: 52 ## objective: 52 ## objective: 52

## objective: 46 ## objective: 46 ## objective: 46

## objective: 44 ## objective: 44

The object we have created is of class ”phylo.to.map” and contains both our optimized tree, the geographic coordinates of our observations, and our underlying base map for plotting.

__________________________________________________________________ tortoise.phymap

## Object of class "phylo.to.map" containing: ## ## (1) A phylogenetic tree with 15 tips and 14 internal nodes. ## ## (2) A geographic map with range: ##      -5.01N, 1.44N ##      -91.67W, -75.22W. ## ## (3) A table containing 15 geographic coordinates (may include ##     more than one set per species). ## ## If optimized, tree nodes have been rotated to maximize alignment ## with the map when the tree is plotted in a rightwards direction.

Finally, we were ready to plot our tree. Here, we must remember to specify the x and y axis limits (via the arguments xlim and ylim, respectively) based on the geographic coordinates of our geolocality data.

__________________________________________________________________ plot(tortoise.phymap,direction="rightwards",pts=FALSE,   xlim=c(-92.25,-89.25),ylim=c(-1.8,0.75),ftype="i",fsize=0.8,   lty="dashed",map.bg="lightgreen",colors="slategrey")

The results can be seen in Fig. 20. Although the base map from mapdata is sufficiently high resolution for our purposes here, higher resolution maps are available for some regions, and it is even possible to import and use a custom map—should we be so inclined (e.g., Quach, Reynolds & Revell, 2019).

Figure 20 A phylogenetic tree of Galapagos tortoises projected onto a geographic map.

Phylogenetic data and geographic locality information are based on Poulakakis et al. (2020). See main text for more details.

Projecting trees into phenotypic space

Along with projecting a phylogenetic tree onto a geographic map (as we just saw), and projecting traits onto the edges and nodes of a plotted tree, the phytools package also contains multiple methods to project a tree into a space defined by our traits. Undoubtedly, the most popular of these are phylomorphospace, which projects a tree into a bivariate quantitative trait space (Sidlauskas, 2008; e.g., Friedman et al., 2016; Martins et al., 2021), and phenogram, which projects a tree into a space defined by time since the root on the horizontal and phenotype on the vertical (typically called a “traitgram,” see Evans et al., 2009; e.g., Martinez et al., 2020; Chazot et al., 2021). We saw how phenogram works in Fig. 14B of any earlier section of this article. Here, I will focus on the phytools method phylomorphospace.

For this example I will be using a time-calibrated phylogeny of 11 vertebrate species from the TimeTree website (Hedges, Dudley & Kumar, 2006), and a phenotypic trait dataset of body mass (in kg) and mean litter size. (The latter dataset was generated from Wikipedia and other sources: i.e., “Googling it.”)

__________________________________________________________________ data(vertebrate.tree) data(vertebrate.data) head(vertebrate.data)

##                           Mass Length Litter_size ## Carcharodon_carcharias 2268.00  6.100        10.0 ## Carassius_auratus         0.91  0.380       200.0 ## Latimeria_chalumnae      80.00  2.000        15.0 ## Iguana_iguana             8.00  2.000        50.5 ## Turdus_migratorius        0.28  0.094         4.0 ## Homo_sapiens             80.00  1.700         1.1

We should see that our data frame actually has three columns–but henceforward I will just use the first and the third of these.

Normally, we could pass our data frame or matrix and phylogeny directly to the phylomorphospace function and obtain a plot. phylomorphospace would then undertake to project the tree, using Maximum Likelihood reconstructed ancestral values for both traits as the positions for internal nodes. In this case, however, I would prefer to first reconstruct ancestral states on a log scale, back-transform my estimated values to the original space, and then use these back-transformed reconstruction as my node positions. Fortunately, phylomorphospace allows that. (In addition to the reasoning I provided in an earlier section, the logic of reconstructing ancestral states on a log scale is because quantitative traits in general, and morphometric data in particular, often satisfy the Brownian motion assumption better on a logarithmic than linear scale. The reasoning of back-transforming before plotting or reporting our results is simply because most human brains, mine included, are more adept at interpreting values on an additive rather than multiplicative scale.)

For the first step, I will use the phytools ancestral state estimation function fastAnc. fastAnc computes Maximum Likelihood ancestral states for one input character vector at a time, so we just need to iterate across the two columns (of interest) in our data frame using an apply call as follows.

__________________________________________________________________ vertebrate.ace<-exp(apply(log(vertebrate.data[,c(1,3)]),   2,fastAnc,tree=vertebrate.tree)) vertebrate.ace

##          Mass Litter_size ## 12  25.571594   15.552422 ## 13  17.834793   16.114126 ## 14  16.827622   14.481309 ## 15   8.801275    8.791375 ## 16  20.362215    1.653955 ## 17  17.335294    1.442396 ## 18  24.604666    1.610416 ## 19 152.078728    1.737334 ## 20 301.219076    1.582780 ## 21   6.329908    9.613237

This gives us a set of reconstructed values on our original (linear) scale, but in which the reconstruction was performed on a log scale, and then back-transformed. Finally, let us create our phylomorphospace plot.

__________________________________________________________________ par(mar=c(5.1,4.1,0.6,2.1)) phylomorphospace(vertebrate.tree,   vertebrate.data[,c(1,3)],A=vertebrate.ace,log="xy",   xlim=c(1e-4,1e6),ylim=c(0.5,200),bty="n",label="off",   axes=FALSE,xlab="Mass (kg)",ylab="Litter size",   node.size=c(0,0)) axis(1,at=10ˆseq(-3,5,by=2),   labels=prettyNum(10ˆseq(-3,5,by=2),big.mark=","),   las=1,cex.axis=0.6) axis(2,at=10ˆseq(0,2,by=1),   labels=prettyNum(10ˆseq(0,2,by=1),big.mark=","),   las=1,cex.axis=0.7) cols<-setNames(RColorBrewer::brewer.pal(   nrow(vertebrate.data),"Paired"),   rownames(vertebrate.data)) points(vertebrate.data[,c(1,3)],pch=21,bg=cols,cex=1.2) ind<-order(rownames(vertebrate.data)) legend("topleft",gsub("_"," ",   rownames(vertebrate.data))[ind],pch=21,pt.cex=1.2,   pt.bg=cols[ind],cex=0.6,bty="n",text.font=3)

Here, I chose to graph the projection without taxon labels, then add different colored points and a legend to put the label information back on the plot (sorting my labels alphabetically as I did this). The result can be seen in Fig. 21. As readers can probably imagine, taxon labels on a phylomorphospace plot can easily become very messy–particularly for larger trees.

Figure 21 Phylomorphospace of body mass and litter size for a selection of vertebrate species.

The underlying phylogenetic tree was obtained from Hedges, Dudley & Kumar (2006). See main text for additional details.

Plotting phenotypic data at the tips of the tree

In addition to projecting phylogenies into trait spaces, and plotting observed or reconstructed trait values on the tree, phytools possesses a number of different plotting methods that can also help us undertake the (at least, conceptually) simple task of visualizing comparative trait data for species at the tips of the tree.

This might be done in various ways. For instance, we could graph the values of a quantitative trait adjacent to the tip labels using a bar or box, or we might plot the presence or absence of different lineages from a habitat type next to the tips of the tree (Revell, 2014b). Numerous such approaches have been developed and implemented in the phytools package, and many of these are shown in my recent book (Revell & Harmon, 2022).

Here, I will illustrate just one such method in which a color gradient is used to visualize trait values for a set of quantitative characters at the tips of the tree (Revell, 2014b). The phytools implementation of this plotting method is called phylo.heatmap, and it has been used in numerous published articles (e.g., Goelen et al., 2020; Hultgren et al., 2021; Molina-Mora et al., 2021; Huang et al., 2022; Morales-Poole et al., 2022). For this example, we will use a phylogenetic tree and log-transformed morphological trait dataset for Anolis lizards from Mahler et al. (2010). To load these data, readers should run the following.

__________________________________________________________________ data(anoletree) data(anole.data) head(anole.data)

##                 SVL      HL     HLL     FLL     LAM      TL ## ahli        4.03913 2.88266 3.96202 3.34498 2.86620 4.50400 ## allogus     4.04014 2.86103 3.94018 3.33829 2.80827 4.52189 ## rubribarbus 4.07847 2.89425 3.96135 3.35641 2.86751 4.56108 ## imias       4.09969 2.85293 3.98565 3.41402 2.94375 4.65242 ## sagrei      4.06716 2.83515 3.85786 3.24267 2.91872 4.77603 ## bremeri     4.11337 2.86044 3.90039 3.30585 2.97009 4.72996

Our trait data object, anole.data, is a data frame with six trait columns for various phylogenetic traits.

We could visualize our data directly; however, the effect of overall size (data column “SVL”) would tend to obscure any interesting patterns of residual variation and covariation in body shape among the species in our tree. As such, primarily in an effort to control for overall size, I will first run a phylogenetic principal components analysis (Revell, 2009) using the phytools function phyl.pca. A phylogenetic principal components analysis (mentioned earlier with respect to the Broeckhoven et al., 2016 dataset) is similar to a regular PCA except that we account for non-independence of the information for different species in our data rotation (Revell, 2009).

__________________________________________________________________ anole.ppca<-phyl.pca(anoletree,anole.data,mode="corr") anole.ppca

## Phylogenetic pca ## Standard deviations: ##       PC1       PC2       PC3       PC4       PC5       PC6 ## 2.2896942 0.6674345 0.4381067 0.2997973 0.1395612 0.1026573 ## Loads: ##            PC1         PC2         PC3         PC4           PC5 ## SVL -0.9782776 -0.01988115  0.14487425 -0.11332244  0.0781070110 ## HL  -0.9736568 -0.03879982  0.13442473 -0.15596460 -0.0852979941 ## HLL -0.9711545  0.14491400  0.02151524  0.17058611 -0.0588208480 ## FLL -0.9759133 -0.02087140  0.14486273  0.14149988  0.0475205990 ## LAM -0.8299594 -0.50437051 -0.23796010  0.01194704  0.0004983465 ## TL  -0.8679195  0.40956428 -0.27350654 -0.05871034  0.0195584629 ##              PC6 ## SVL -0.051442939 ## HL   0.028570939 ## HLL -0.053257988 ## FLL  0.062386141 ## LAM -0.003133966 ## TL   0.018373275

Our PC loadings show us the the first principal component dimension is strongly negatively correlated with all of the traits in our analysis. We could consider this the “size” axis. Principal component 2 is most strongly (negatively) correlated with the character “LAM”, number of adhesive toepad scales called lamellae; and most positively correlated with “TL”, tail length.

Let us compute the principal component scores for all of our species.

__________________________________________________________________ anole.pc scores<-scores(anole.ppca) head(anole.pc_scores)

##                    PC1       PC2         PC3       PC4         PC5 ## ahli        -0.1747576 0.8697064  1.52379491 1.6029659 -0.23955421 ## allogus      0.1646585 1.4017806  1.74506491 1.5358005 -0.08089546 ## rubribarbus -0.4925001 1.0413268  1.45866163 1.4180850 -0.05716104 ## imias       -1.1608049 0.7514380  0.75822327 1.7127381  0.35013533 ## sagrei      -0.3486332 1.2632997 -0.05102313 0.7317455  0.37217463 ## bremeri     -0.9714818 0.6943196  0.11689334 0.9290039  0.43486041 ##                    PC6 ## ahli        -0.1626049 ## allogus     -0.1245855 ## rubribarbus -0.1797060 ## imias       -0.1340347 ## sagrei      -0.1484157 ## bremeri     -0.2304513

Since the sign of each principal component is arbitrary (principal components are vectors), we will now “flip” the sign of PC 1 –so that it switches from negative size to simply “size.”

__________________________________________________________________ anole.pc scores[,1]<--anole.pc scores[,1]

Finally, let us graph our results using the phylo.heatmap function. Seeing as the variance in our different principal component dimensions are quite different from PC to PC, we will standardize them to have a constant variance using standardize=TRUE. As we have done in other exercises of this article, we can update the default color palette of the plot using the argument colors. Here, I will use the colorblind-friendly viridis color palette from the viridisLite package (Garnier et al., 2022) that we learned about earlier.

__________________________________________________________________ phylo.heatmap(anoletree,anole.pc scores,   standardize=TRUE,fsize=c(0.4,0.7,0.7),pts=FALSE,   split=c(0.6,0.4),colors=viridisLite::viridis(n=40,   direction=-1),mar=rep(0.1,4))

In Fig. 22 we can already begin to discern some of the interesting ecomorphological phenotypic patterns of Anolis lizards (Losos, 2009). For instance, the largest species (known as “crown-giants,” Losos, 2009), that is, those species with the highest values of PC 1, tend to have moderate or low values for PC 2: meaning they have larger lamellae and shorter tails, controlling for their body size. By contrast some of the smallest species (on PC 1) have among the highest values for PC 2 (tail length). These are the “grass-bush” anoles that perch on grass and bushes near the grown, and use their long tails to control body pitch while jumping. We can likewise see that this combination of phenotypic traits (large body size and large lamellae; small body size and long tail) has evolved independently in different parts of the phylogenetic tree. Just by visualizing our data and learning this, we are already doing phylogenetic comparative methods. Neat!

Figure 22 Phylogenetic heatmap showing principal components from a phylogenetic PCA of six morphological traits of Anolis lizards.

Tree and data are from Mahler et al. (2010). See main text for additional details.

Relationship of phytools to other packages

The phytools package has grown to become (along with ape, phangorn, and geiger) among the most important core packages for phylogenetic analysis in the R environment. As of the time of writing, the original publication describing phytools (Revell, 2012) had been cited more than 7,300 times on Google Scholar and continues to be cited over 1,000 times per year.

In many respect, however, phytools owes its existence to a number of other packages making up the R phylogenetics ecosytem and from which it imports crtical functionality. In particular, phytools depends on the object classes and methods of the core R phylogenetics package, ape (Paradis, Claude & Strimmer, 2004; Popescu, Huber & Paradis, 2012; Paradis & Schliep, 2019). In addition, phytools relies on a number of different methods from the multifunctional phylogenetic inference package, phangorn (Schliep, 2011). Finally, phytools is designed to interact with a variety of other function R phylogenetics libraries, especially the geiger package (Harmon et al., 2008; Pennell et al., 2014), which phytools “suggests” but does not import.

Outside of the phylogenetics ecosystem, phytools also presently depends on or imports from a number of other R packages including clusterGeneration (Qiu & Joe, 2023), coda (Plummer et al., 2006), combinat (Chasalow, 2012), doParallel (Microsoft Corporation & Weston, 2022a), expm (Maechler, Dutang & Goulet, 2023), foreach (Microsoft Corporation & Weston, 2022b), maps (Becker et al., 2022a), MASS (Venables & Ripley, 2002), mnormt (Azzalini & Genz, 2022), nlme (Pinheiro & Bates, 2000; Pinheiro, Bates & R Core Team, 2022), numDeriv (Gilbert & Varadhan, 2019), optimParallel (Gerber & Furrer, 2019; Lemon, 2006), and scatterplot3d (Ligges & Mächler, 2003), although dependency relationships are dynamic as packages evolve and may change.

Conclusions

More than a decade has passed since the original and only article describing phytools was published (Revell, 2012). Since that time, the phytools package has both evolved into one of the core function libraries of the R phylogenetics ecosystem, and expanded manyfold in size and scope. As such, I decided the literature reference for phytools was sorely in need of updating. In creating one, however, I was determined to make something that could serve as more than a placeholder to capture citations of the phytools package. I hope that what I have provided here will help guide some new phytools users towards interesting analytical tools, as well as perhaps inspire experienced phytools and R phylogenetics researchers to generate new types of questions and data that will in turn help motivate continued development of the phytools package into the future.

Software and data availability

The phytools R package is free and open source, and can be downloaded from its CRAN (https://CRAN.R-project.org/package=phytools) or GitHub (https://github.com/liamrevell/phytools) pages. More information about the phytools package can be obtained from the software documentation pages, my phytools blog (http://blog.phytools.org), or via my recent book with Luke Harmon (Revell & Harmon, 2022).

This article was written in Rmarkdown (Xie, Allaire & Grolemund, 2018; Xie, Dervieux & Riederer, 2020; Allaire et al., 2023), and developed with the help of both bookdown (Xie, 2016; Xie, 2023) and the posit Rstudio IDE (RStudio Team, 2020). All data used in the analyses of this article are packaged with the phytools R library versions on CRAN and GitHub (links above). Markdown code necessary to exactly rebuild the submitted version of this article (including its analyses and figures) are available at https://doi.org/10.5281/zenodo.10067375. A previous version of this article was posted to the preprint server bioRxiv (https://doi.org/10.1101/2023.03.08.531791).

Additional Information and Declarations

Competing Interests

Author Contributions

Data Availability

The author declares that he has no competing interests.

Liam J. Revell conceived and designed the experiments, performed the experiments, analyzed the data, prepared figures and/or tables, authored or reviewed drafts of the article, and approved the final draft.

The following information was supplied regarding data availability:

The markdown code & data files necessary to exactly reproduce the analyses & figures of this article are available at GitHub and Zenodo:

- https://github.com/liamrevell/Revell.phytools-v2/

- https://doi.org/10.5281/zenodo.10067376.

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
