# Peer review of "phytools 2.0: an updated R ecosystem for phylogenetic comparative methods (and other things)"

_PeerJ, doi:10.7717/peerj.16505_

## Round 0.1 · original submission · Minor Revisions

I agree with the reviewers that this is a much welcome and timely update on an important software package in the phylogenetic comparative methods ecosystem. The reviewers make a number of insightful and constructive comments for improvement. I'll add the following suggestions, mostly regarding the code itself.

1. Open Source licenses are not all equal, and therefor in Software and Data Availability, I suggest to give the actual license (GPLv2+, based on what the DESCRIPTION file says?). Given that there's no LICENSE file in the repository, I would also suggest to add that information to the README file there, so visitors don't have to guess (or know to look into the DESCRIPTION file).

2. I assume that for reproducing the manuscript's computations and figures, the author is referring not to the root directory in the liamrevell/Revell.phytools-v2 GitHub repository, but to the "peerj" subdirectory there? I suggest making this clear in the manuscript (Software and Data Availability section). Also, this repository seems to lack a license, and thus one needs to be assigned.

4. To rerun the Rmarkdown source in this directory, upgrading phytools to the latest release on CRAN turns out to be insufficient. (It results in version 1.5-1, which for example lacks the butterfly dataset.) Upgrading instead to the GitHub repository's version results in a version considerably higher (1.7-7) than the one reported in the manuscript. It is thus a little unclear what the author's CRAN release policy is. It seems at least it would help if the Rmarkdown included a minimum version check. If the author's CRAN release policy is indeed to make much less frequent CRAN releases than version increments in the GitHub repository, then this seems worth noting (at least in the README of the repo).

5. There is a bug in the data file for betaCoV.tree, which I posted as issue #131 on the phytools Github repository. (This bug causes the Rmarkdown source of the manuscript to fail rendering at line 971 -- even though the actual problem is in line 956 --; perhaps the author uses a filesystem with case-insensitive filenames?)

·

Basic reporting

This paper, which describes major updates to the highly used and highly cited (and probably more used than cited), R package phytools. This package has been under continuous development over the last decade, with new features added regularly. The paper then is a combination of "status update" and "tutorial". It is written in the informal style typical of a blog, rather than that of a conventional paper; I am honestly fine with this -- it is clearly written and references the literature where appropriate. I note that there is a lot of overlap between the material presented here and that which appears in the author's recent (excellent) book (Revell and Harmon 2022, Princeton). The author addresses this overlap, and I will leave it to the judgement of the editor as to whether this is problematic.

Overall, this is a useful publication documenting the evolution of a very useful R package. The phylogenetics community is indebted to the author for implementing so many various methods into an easy-to-use and well-documented piece of software and thereby facilitating so many analyses that would have been done otherwise.

Experimental design

I appreciated the detailed walkthrough. I think it is helpful to have all the little bits (e.g., the plot aesthetics, etc.) spelled out explicitly rather than left as an exercise to the reader. The code all runs (see caveat mentioned in 4) and the analyses work as advertised. I also appreciate that both the codebase and the article itself are version controlled.

Validity of the findings

I understand that the gamma test is technically done and described correctly but in the literature: 1) the null model is almost always a CR birth-death model and not a pure-birth model; 2) when the null is a CR birth-death, this is almost always considered a one-sided test (i.e., only values that are significantly negative are of interest). Of course, I understand that this is beside the core point that is being made; nonetheless, I think this introduces some unnecessary confusion about the use of this statistic.

Additional comments

There is some inconsistency in the naming of objects -- both "." and "_" are used throughout. I would advise that these be standardized. Related to this, I encountered a strange problem (and perhaps this is on my end that when I copied and pasted from the PDF into RStudio (on Mac), the "_" character was not recognized and the code didn't run. I ran the code directly from the GitHub page and everything worked fine.

Reviewer 2 ·

Basic reporting

This paper meets all the basic reporting requirements. Although one might argue language is not entirely professional more conversational in tone but it makes it more approachable.

Experimental design

NA this is a software description

Validity of the findings

NA this is a software description

Additional comments

I agree that a new paper describing phytools is timely, as there have been many additions since the 2012 paper including several important and exciting developments highlighted in this paper. The new book by Revell and Harmon is a comprehensive overview of PCM and as such there is still a need for a paper focused on phytools. Although an unusual form for a software description, it is useful, if a little self-indulgent at times. It is essentially a curated collection of vignettes focusing on the more recently implemented methods with some general information that links them together. The sections: discrete traits, continuous characters, diversification, and visualization provides a very clear general structure. Although, you could consider removing the diversification section, as there are other packages that provide much greater functionality for exploring speciation and extinction and I am not sure the functions in phytools provides novel methods. While the target audience is both new and returning users, my recommended changes focus on the likely needs of new users. I have thus suggested where additional explanations may be necessary to make this paper as helpful as possible to someone just getting started.

I appreciate it is a hard balance between being exhaustive in the list of tools and providing examples and explanation, as well as between providing too much detail and not enough. However, if your target audience includes new users then I think some details are missing. For example, folks would very much appreciate a table of the major functions for each section. These additions shouldn’t add to the length of the paper, as you can cut out some of the unnecessary aspects of the code, such as breaking halfway through an analysis to examine the output and judiciously editing the more expansive, conversational-style language.

Most software descriptions include context and comparisons to other available software and discussions of limitations. Phytools is a well-established, widely used, and extensive package and so I understand the wish and need to focus on its implementations. However, I think readers would benefit from some additional context and discussion of limitations of the methods as well as alternatives.

Structure and language, there are several parenthetical paragraphs which I recommend changing, as well as some odd paragraph transitions and figures that could be placed closer to code provided to generate them. It might also be worth considering bolding the warnings you provide e.g. lines 780-781

Note: I did not run the code examples to check that these worked as expected.


Line 34: “Modern phylogenetic comparative methods are not new” – this is picky but recommend changing to phylogenetic comparative methods are not new, I think most would agree that PCM were catalyzed by Felsenstein 1985 but I am not sure it should be referred to as modern, which means present or recent past 40 years is a long time!

Lines 84-104: I am not sure so much space needs to be devoted to installing, if you are assuming some familiarity with R just tell folks to download from CRAN, as this is a very basic R task.

Lines 105-118: the list of methods and models that you have chosen to highlight would be easier to read in a table and I suggest adding a column with a brief description of the use of each, so that new users don’t have to look up for example what an extended Markov model is used for.

Line 119: if part of your target audience is new users it might be helpful to provide a description of what stochastic mapping is for e.g. “Reconstructing the history of a discrete character with stochastic character mapping”. This wouldn’t be necessary if you chose to add the table suggested above.

Lines 129-133: it is most unusual to have a whole parenthetical paragraph and I think it confusing. Perhaps help readers by saying there are two popular ways to implement stochastic character mapping either first fitting a character model using ML or to sample using MCMC. Readers might also appreciate if you explain what they might want to consider when choosing one method over the other. I also think it would be helpful to provide context, especially comparing it to SFREEMAP also available in R.

Lines 138-139: I also would recommend being explicit about whether the example you provide fits one of the aforementioned methods or is distinct – again it will really help readers not familiar with these methods.

Lines 151-155: another parenthetical paragraph = confusing. I don’t think you need the parentheses you start “in this example” so readers are aware you are discussing something specific to the example and it is not a general topic. I do like the inclusion of this point, as for new users the differences between characters, factors etc. can be very confusing.

Lines 197-200: recommend moving the parenthetical sentences to below the example code, as it will help emphasize the code implements the weighted analysis not the alternatives.

Line 201: why use 1000 simulations, how does the user know how many simulations to run to get a reliable results?

Lines 211-215: either explain why the added complexity of the viridis palette is useful i.e. improves graph readability especially for folks with colorblindness or perhaps make it simpler and just pass it some colors.

Lines 220 -233: “perhaps most often users…” and “users undertaking a stochastic character mapping..”, phrasing is not very helpful just because it is done doesn’t mean it should be done. Help the reader understand that when looking at maps it is useful to a and b as you learn x and y.

Figure 3 – recommend a single graph with the two transition rates overlaid, as it is easier to compare the distributions on the same graph.

Line 271 – before moving on to the next topic I think it is important to add a brief discussion of the benefits/limitations to the methods implemented. Can one account for variation in tree structure for the analyses and visualization? Can one account for rate variation across the phylogeny? Can one build density maps when there are more than two states? Can you include polymorphism?

Line 271 – good choice to highlight this new model! But perhaps it should be first as this is a new model of trait evolution and then you discuss mapping as a tool that uses these models. You can briefly discuss the available models for discrete traits including ones not mentioned currently then segue into the new model. This would be a little more comprehensive, which would be helpful for new users. You could move the mapping of a polyMk to the second section on character mapping

Line 377-382 – not really necessary to look at the results halfway through.

Figure 6 – move to after line 409, so that it is closer to the code used to generate it.

Line 423 – why use base color graphics when you have already introduced viridis, an explanation for the choice is useful especially as you say it is difficult for you to read, surely you don’t want to encourage others to generate graphics that are difficult for folks with colorblindness to read?

Line 430 – why not move the LTT plot to the relevant later section? Also, I think it might be important to highlight the info given at the end in parentheses at the beginning - this plot is only meaningful with complete taxon sampling.

Line 438 – before moving on to the next topic I think it is important to add a brief discussion of the benefits/limitations to the methods implemented. What are the benefits of a polymorphic model vs a regular Mk model where the polymorphic states are code as separate states? Can one account for rate variation across the phylogeny? Can one account for variation in tree structure for the analyses and visualization?

Lines 439-451: the list of methods and models that you have chosen to highlight would be easier to read in a table and I suggest adding a column with a brief description of the use of each.

Line 515: I think it would be very helpful for many readers to expand on this point, to explicitly explain why the different procedures can provide very different conclusions sometimes.

Lines 566-569: if you are going to explain them do it fully – it is not clear which parameter $a and $y etc are..

Line 572: this is all very well to say that the percentage of burn-in can be changed but many readers may not know why this is important and even if they do, how they go about checking to see if 20% is sufficient. It would be helpful to provide this information.

Line 521: move figure 9 here out of the Bayesian ASR section.

Line 620: before moving on to the next topic I think it is important to add a brief discussion of the benefits/limitations to the methods implemented. Can one account for variation in tree structure for the analyses and visualization? Can you include rate variation?

Lines 621-622: it might be useful to point readers to other packages that contain additional multivariate methods here.

Figure 12: move to line 654 to follow the code for plotting it

Line 672: provide brief explanation of the different models here, the reader will be confused if they can’t interpret the results and have to search the out from tropidurid.fits to understand them.

Lines 694-719: I am not sure this is necessary as they are not discussed in the text.

Line 720: before moving on to the next topic I think it is important to add a brief discussion of the benefits/limitations to the methods implemented or alternatives. For example, how does this compare to the models used to identify integration and modularity?

Line 720: similar to the discrete section, it might work better to highlight this new model of trait evolution first and put it into context with the other models of continuous trait evolution at the beginning then move on to methods that implement these models. For readers new to these methods, briefly explaining the basic models first would be helpful – this can also include OU, which is an important model not mentioned currently and would allow phylogenetic signal to be explained in the context of Brownian motion.

Lines 747-753: it is hard to get a feel for what is biologically realistic here and I suspect it might depend on the taxonomic scope of the analysis. Are these parameter values 0.1 and 10 really bounding what is feasible or is it far too broad or not expansive enough? How does one determine this? Given this is a relatively new method in phytools, some more guidance would be helpful.

Line 760: projection not project

Lines 765-767: move to the explanation of how visual inspection will help to line 761 before the explanation of the type of plot.

Figure 13 move to just before line 768.

Figure 14 move to just before line 783

Line 787: before moving on to the next topic I think it is important to add a brief discussion of the benefits/limitations to the methods implemented or alternatives. For example, how does this compare to the variable rates model in BayesTraits or RevBayes?

Lines 788-920: Given there aren’t that many methods for diversification in phytools I suggest you consider excluding this section, as the functions available are not unqiue and there are several other packages that provide more functionality for modelling speciation and extinction.

Lines 924-932: this paragraph seems unnecessary; just say in this final section I’ll illustrate a few popular plotting methods not covered in the preceding sections.

Line 921: again, readers I think would find a table listing the various visualization options very helpful.

Line 947: this shouldn’t be a new paragraph.

Figure 18: should be just before line 981 putting it next to the code used to plot it.

Line 980: why use RColorBrewer over the others you have already introduced like viridis? It seems unnecessary complexity and will potentially add confusion for new users.

Line 990: instead of focusing on how laborious it was, perhaps state the type of data which is needed – lat/long coordinates.

Line 1050: why use randomcoloR over the others you have already introduced like viridis? It seems unnecessary complexity and will potentially add confusion for new users. Here you felt the need to explain how to download it but not for RColorBrewer – why?

Line 1055: how would one restrict the map? Seems like a useful tidbit to include as many folks won’t have a global dataset.

Lines 1078-1083: explain why you prefer to reconstruct ASR on log scale and then back-transform to original space.

Line 1117: it looks like all the data are logged already? Perhaps w

·

Basic reporting

I have evaluated the basic reporting requirements and the article appears to meet or exceed the indicated standards.

Experimental design

I have evaluated the experimental design requirements and the article appears to meet or exceed the indicated standards.

Validity of the findings

I have evaluated the validity requirements and the article appears to meet or exceed the indicated standards.

Additional comments

Review for:
phytools 2.0: An updated R ecosystem for phylogenetic comparative methods (and other things)

Overall, I have evaluated the PeerJ reporting requirements, and the article appears to meet or exceed the indicated standards in all areas. I have a few general comments that may improve a few areas of the text if the author chooses to implement them.

First:
I have no concerns about the technical implementation of the provided examples. However, I wonder if the content about model averaging is presented in the best way. I am unsure if there is consensus within the field of systematic biology that model-averaging applied in this context (e.g., starting around line 205) can be considered a "best practice." I do not think the author means to imply that it is -- however, I think a novice reader may read it this way. Many students getting into PCMs do not understand what they are doing and follow example tutorials without thinking too much. I have no statistical argument against model averaging; I think it makes sense in some contexts. However, it is unclear to me whether it makes sense to average the results across models that make incompatible statements about the data-generating process (e.g., ER and irreversible models), even if those model outputs are weighted according to their model uncertainty. The fact that phytools 2.0 can do this notwithstanding, I suggest the author reconsider how a naive reader may interpret the presentation of this example. If, on the other hand, the author believes that this *is* a preferred way to account for model-fitting uncertainty, I think it may be worth being a bit clearer on that point and perhaps providing some context as to when this is advisable over standard fitting and model comparison procedures.

In any case, a naive reader may interpret the provided example as a suggestion of an optimal workflow rather than a possible workflow and end up with results that are difficult to interpret (e.g., the bimodality in Figure 3). [an aside: I think the bimodality in the posterior estimates for the number of changes may break the HPD estimator because these are no longer posterior estimates under a given model (indeed, what is the "model" when the parameter estimates are model-averaged?) -- I also think there may need to be 2 HPD intervals in the case of some bimodal posterior distributions (e.g., in some cases, the HPD would no longer be similar to to the 2.5 and 97.5 quantiles)].

Second:
I think it may be helpful to include an example of examining the likelihood surface for the discrete and/or continuous character models (like Fig. 9). Does phytools provide tools for visualizing model convergence (e.g., difficulty noted on lines 401-410) for these models? Perhaps this should be part of a "good" workflow, even if such diagnostics are rarely reported in the literature (I note this only because exemplifying a "good" workflow seems to be a goal of this paper). Perhaps the section after the polymorphic character model would be a suitable place for such an example, as the author already notes the difficulty in identifying the ML solution for these models (e.g., such as Fig 17).

A minor detail:
622-629 -- mvMORPH, RPANDA, bayou, SLOUCH, PCMFit, and PhylogeneticEM have sophisticated machinery for fitting multivariate models -- perhaps worth mentioning some of these. For example, mvMORPH can fit multi-regime models in which the VCV can vary across specified regimes and/or according to different models (e.g., BM, OU, etc.). Another minor and related point that could be clarified in this example is the underlying model assumption (e.g., this specific example assumes mvBM).

---

## Round 0.2 · accepted · Accept

I appreciate the very thorough response to my and all reviewers' comments and find that all concerns and suggestions have been sufficiently well addressed.

I also reviewed the revised manuscript and found a number of typos and similar nature. I believe these do not warrant another minor revision and will therefore pass them to the PeerJ staff for dealing with during the galley proof stage.

Examples:

Line 15-16: "an important research tool": the tracked changes and the cover abstract have "tools" (plural) but perhaps this was subsequently corrected?
Line 20: "as well numerous" should be "as well as numerous"?
Line 65: "not restricted to, but especially, visualization" I'd suggest to switch the order: "especially, but not restricted to, visualization"
Line 231: "each differs one from the other" should be "each one differs from the other"?
Line 251-252: "these values marginal posterior probabilities" should be "these marginal posterior probabilities values"?
Line 298: "also" duplicated
Line 503: "important aspect of this model it allows" should be "important aspect of this model is that it allows"
Line 805: "are among the most lightly all cordylid lizards" should be "are among the most lightly armored of all cordylid lizards"?
Line 886: "heirarchical" typo (should be "hierarchical")
Line 989: "a simple projection our phylogeny" should be "a simple projection of our phylogeny"?
Line 1017: "Various additional R package": "package" should be in plural
Line 1019: "Haegeman 2023), and others": using both "including" and "and others" is redundant
Line 1078: "we can more easily visualization" should be "we can more easily visualize"
Line 1091: "know elapids" should be "known elapids"
Line 1474: "Outside phylogenetics as such" I had to read this 3x times to get it (perhaps just me); possibly use "Outside of the phylogenetics ecosystem".
Figure 21 caption: "uderlying phylogenetic tree" typo, should be "underlying phylogenetic tree"